# Never Give Up: Learning Directed Exploration Strategies

**Adrià Puigdomènech Badia**[*]   **Pablo Sprechmann**[*]   **Alex Vitvitskyi**   **Daniel Guo**

**Bilal Piot**   **Steven Kapturowski**   **Olivier Tieleman**   **Martín Arjovsky**

**Alexander Pritzel**   **Andew Bolt**   **Charles Blundell**

DeepMind `{adriap, psprechmann, avlife, danielguo,`
`        piot, skapturowski, tieleman,`
`        apritzel, abolt, cblundell}@google.com`

## Abstract

We propose a reinforcement learning agent to solve hard exploration games by learning a range of directed exploratory policies. We construct an episodic memory-based intrinsic reward using k-nearest neighbors over the agent's recent experience to train the directed exploratory policies, thereby encouraging the agent to repeatedly revisit all states in its environment. A self-supervised inverse dynamics model is used to train the embeddings of the nearest neighbour lookup, biasing the novelty signal towards what the agent can control. We employ the framework of Universal Value Function Approximators (UVFA) to simultaneously learn many directed exploration policies with the same neural network, with different trade-offs between exploration and exploitation. By using the same neural network for different degrees of exploration/exploitation, transfer is demonstrated from predominantly exploratory policies yielding effective exploitative policies. The proposed method can be incorporated to run with modern distributed RL agents that collect large amounts of experience from many actors running in parallel on separate environment instances. Our method doubles the performance of the base agent in all hard exploration in the Atari-57 suite while maintaining a very high score across the remaining games, obtaining a median human normalised score of 1344.0%. Notably, the proposed method is the first algorithm to achieve non-zero rewards (with a mean score of 8,400) in the game of *Pitfall!* without using demonstrations or hand-crafted features.

## 1 Introduction

The problem of exploration remains one of the major challenges in deep reinforcement learning. In general, methods that guarantee finding an optimal policy require the number of visits to each state–action pair to approach infinity. Strategies that become greedy after a finite number of steps may never learn to act optimally; they may converge prematurely to suboptimal policies, and never gather the data they need to learn. Ensuring that all state-action pairs are encountered infinitely often is the general problem of maintaining exploration (François-Lavet et al., 2018; Sutton & Barto, 2018). The simplest approach for tackling this problem is to consider stochastic policies with a non-zero probability of selecting all actions in each state, e.g. $\epsilon$-greedy or Boltzmann exploration. While these techniques will eventually learn the optimal policy in the tabular setting, they are very inefficient and the steps they require grow exponentially with the size of the state space. Despite these shortcomings, they can perform remarkably well in dense reward scenarios (Mnih et al., 2015). In sparse reward settings, however, they can completely fail to learn, as temporally-extended exploration (also called deep exploration) is crucial to even find the very few rewarding states (Osband et al., 2016).

---

[*]Equal contribution.

Recent approaches have proposed to provide intrinsic rewards to agents to drive exploration, with a focus on demonstrating performance in non-tabular settings. These intrinsic rewards are proportional to some notion of saliency quantifying how different the current state is from those already visited (Bellemare et al., 2016; Haber et al., 2018; Houthooft et al., 2016; Oh et al., 2015; Ostrovski et al., 2017; Pathak et al., 2017; Stadie et al., 2015). As the agent explores the environment and becomes familiar with it, the exploration bonus disappears and learning is only driven by extrinsic rewards. This is a sensible idea as the goal is to maximise the expected sum of extrinsic rewards. While very good results have been achieved on some very hard exploration tasks, these algorithms face a fundamental limitation: after the novelty of a state has vanished, the agent is not encouraged to visit it again, regardless of the downstream learning opportunities it might allow (Bellemare et al., 2016; Ecoffet et al., 2019; Stanton & Clune, 2018). Other methods estimate predictive forward models (Haber et al., 2018; Houthooft et al., 2016; Oh et al., 2015; Pathak et al., 2017; Stadie et al., 2015) and use the prediction error as the intrinsic motivation. Explicitly building models like this, particularly from observations, is expensive, error prone, and can be difficult to generalize to arbitrary environments. In the absence of the novelty signal, these algorithms reduce to undirected exploration schemes, maintaining exploration in a non-scalable way. To overcome this problem, a careful calibration between the speed of the learning algorithm and that of the vanishing rewards is required (Ecoffet et al., 2019; Ostrovski et al., 2017).

The main idea of our proposed approach is to jointly learn separate exploration and exploitation policies derived from the same network, in such a way that the exploitative policy can concentrate on maximising the extrinsic reward (solving the task at hand) while the exploratory ones can maintain exploration without eventually reducing to an undirected policy. We propose to jointly learn a family of policies, parametrised using the UVFA framework (Schaul et al., 2015a), with various degrees of exploratory behaviour. The learning of the exploratory policies can be thought of as a set of auxiliary tasks that can help build a shared architecture that continues to develop even in the absence of extrinsic rewards (Jaderberg et al., 2016). We use reinforcement learning to approximate the optimal value function corresponding to several different weightings of intrinsic rewards.

We propose an intrinsic reward that combines per-episode and life-long novelty to explicitly encourage the agent to repeatedly visit all controllable states in the environment over an episode. Episodic novelty encourages an agent to periodically revisit familiar (but potentially not fully explored) states over several episodes, but not within the same episode. Life-long novelty gradually down-modulates states that become progressively more familiar across many episodes. Our episodic novelty uses an episodic memory filled with all previously visited states, encoded using the self-supervised objective of Pathak et al. (2017) to avoid uncontrollable parts of the state space. Episodic novelty is then defined as similarity of the current state to previously stored states. This allows the episodic novelty to rapidly adapt within an episode: every observation made by the agent potentially changes the per-episode novelty significantly. Our life-long novelty multiplicatively modulates the episodic similarity signal and is driven by a Random Network Distillation error (Burda et al., 2018b). In contrast to the episodic novelty, the life-long novelty changes slowly, relying upon gradient descent optimisation (as opposed to an episodic memory write for episodic novelty). Thus, this combined notion of novelty is able to generalize in complex tasks with large, high dimensional state spaces in which a given state is never observed twice, and maintain consistent exploration both within an episode and across episodes.

This paper makes the following contributions: *(i)* defining an exploration bonus combining life-long and episodic novelty to learn exploratory strategies that can maintain exploration throughout the agent's training process (to *never give up*), *(ii)* to learn a family of policies that separate exploration and exploitation using a conditional architecture with shared weights, *(iii)* experimental evidence that the proposed method is scalable and performs on par or better than state-of-the-art methods on hard exploration tasks. Our work differs from Savinov et al. (2018) in that it is not specialised to navigation tasks, our method incorporates a long-term intrinsic reward and is able to separate exploration and exploitation policies. Unlike Stanton & Clune (2018), our work relies on no privileged information and combines both episodic and non-episodic novelty, obtaining superior results. Our work differs from Beyer et al. (2019) in that we learn multiple policies by sharing weights, rather than just a common replay buffer, and our method does not require exact counts and so can scale to more realistic domains such as Atari. The paper is organized as follows. In Section 2 we describe the proposed intrinsic reward. In Section 3, we describe the proposed agent and general framework. In Section 4 we present experimental evaluation.

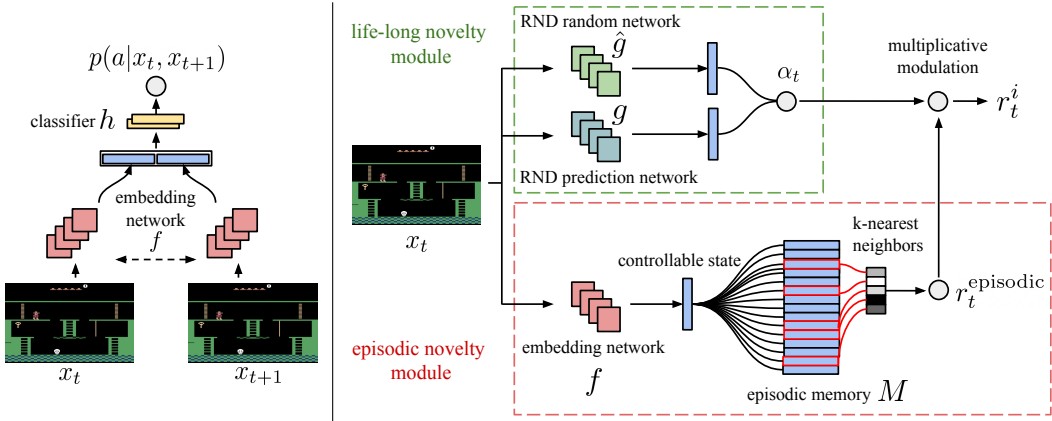

Figure 1: (left) Training architecture for the embedding network (right) NGU's reward generator.

## 2 THE NEVER-GIVE-UP INTRINSIC REWARD

We follow the literature on curiosity-driven exploration, where the extrinsic reward is augmented with an intrinsic reward (or exploration bonus). The augmented reward at time $t$ is then defined as $r_t = r_t^e + \beta r_t^i$, where $r_t^e$ and $r_t^i$ are respectively the extrinsic and intrinsic rewards, and $\beta$ is a positive scalar weighting the relevance of the latter. Deep RL agents are typically trained on the augmented reward $r_t$, while performance is measured on extrinsic reward $r_t^e$ only. This section describes the proposed intrinsic reward $r_t^i$.

Our intrinsic reward $r_t^i$ satisfies three properties: *(i)* it rapidly discourages revisiting the same state within the same episode, *(ii)* it slowly discourages visits to states visited many times across episodes, *(iii)* the notion of state ignores aspects of an environment that are not influenced by an agent's actions.

We begin by providing a general overview of the computation of the proposed intrinsic reward. Then we provide the details of each one of the components. The reward is composed of two blocks: an *episodic novelty module* and an (optional) *life-long novelty module*, represented in red and green respectively in Fig. 1 (right). The episodic novelty module computes our episodic intrinsic reward and is composed of an episodic memory, $M$, and an embedding function $f$, mapping the current observation to a learned representation that we refer to as controllable state. At the beginning of each episode, the episodic memory starts completely empty. At every step, the agent computes an episodic intrinsic reward, $r_t^{\text{episodic}}$, and appends the controllable state corresponding to the current observation to the memory $M$. To determine the bonus, the current observation is compared to the content of the episodic memory $M$. Larger differences produce larger episodic intrinsic rewards. The episodic intrinsic reward $r_t^{\text{episodic}}$ promotes the agent to visit as many different states as possible within a single episode. This means that the notion of novelty ignores inter-episode interactions: a state that has been visited thousands of times gives the same intrinsic reward as a completely new state as long as they are equally novel given the history of the current episode.

A life-long (or inter-episodic) novelty module provides a long-term novelty signal to statefully control the amount of exploration across episodes. We do so by multiplicatively modulating the exploration bonus $r_t^{\text{episodic}}$ with a life-long curiosity factor, $\alpha_t$. Note that this modulation will vanish over time, reducing our method to using the non-modulated reward. Specifically, we combine $\alpha_t$ with $r_t^{\text{episodic}}$ as follows (see also Fig. 1 (right)):

$$r_t^i = r_t^{\text{episodic}} \cdot \min \left\{ \max \left\{ \alpha_t, 1 \right\}, L \right\} \tag{1}$$

where $L$ is a chosen maximum reward scaling (we fix $L = 5$ for all our experiments). Mixing rewards this way, we leverage the long-term novelty detection that $\alpha_t$ offers, while $r_t^i$ continues to encourage our agent to explore all the controllable states.

**Embedding network:** $f : \mathcal{O} \to \mathbb{R}^p$ maps the current observation to a $p$-dimensional vector corresponding to its controllable state. Consider an environment that has a lot of variability independent of the agent's actions, such as navigating a busy city with many pedestrians and vehicles. An agent could

visit a large number of different states (collecting large cumulative intrinsic rewards) without taking any actions. This would not lead to performing any meaningful form of exploration. To avoid such meaningless exploration, given two consecutive observations, we train a Siamese network (Bromley et al., 1994; Koch et al., 2015) $f$ to predict the action taken by the agent to go from one observation to the next (Pathak et al., 2017). Intuitively, all the variability in the environment that is not affected by the action taken by the agent would not be useful to make this prediction. More formally, given a triplet $\{x_t, a_t, x_{t+1}\}$ composed of two consecutive observations, $x_t$ and $x_{t+1}$, and the action taken by the agent $a_t$, we parameterise the conditional likelihood as $p(a|x_t, x_{t+1}) = h(f(x_t), f(x_{t+1}))$, where $h$ is a one hidden layer MLP followed by a softmax. The parameters of both $h$ and $f$ are trained via maximum likelihood. This architecture can be thought of as a Siamese network with a one-layer classifier on top, see Fig. 1 (left) for an illustration. For more details about the architecture, see App. H.1, and hyperparameters, see App. F.

**Episodic memory and intrinsic reward:** The episodic memory $M$ is a dynamically-sized slot-based memory that stores the controllable states in an online fashion (Pritzel et al., 2017). At time $t$, the memory contains the controllable states of all the observations visited in the current episode, $\{f(x_0), f(x_1), \ldots, f(x_{t-1})\}$. Inspired by theoretically-justified exploration methods turning state-action counts into a bonus reward (Strehl & Littman, 2008), we define our intrinsic reward as

$$r_t^{\text{episodic}} = \frac{1}{\sqrt{n(f(x_t))}} \approx \frac{1}{\sqrt{\sum_{f_i \in N_k} K(f(x_t), f_i) + c}} \tag{2}$$

where $n(f(x_t))$ is the counts for the visits to the abstract state $f(x_t)$. We approximate these counts $n(f(x_t))$ as the sum of the similarities given by a kernel function $K : \mathbb{R}^p \times \mathbb{R}^p \to \mathbb{R}$, over the content of $M$. In practice, pseudo-counts are computed using the $k$-nearest neighbors of $f(x_t)$ in the memory $M$, denoted by $N_k = \{f_i\}_{i=1}^k$. The constant $c$ guarantees a minimum amount of "pseudo-counts" (fixed to 0.001 in all our experiments). Note that when $K$ is a Dirac delta function, the approximation becomes exact but consequently provides no generalisation of exploration required for very large state spaces. Following Blundell et al. (2016); Pritzel et al. (2017), we use the inverse kernel for $K$,

$$K(x, y) = \frac{\epsilon}{\frac{d^2(x,y)}{d_m^2} + \epsilon} \tag{3}$$

where $\epsilon$ is a small constant (fixed to $10^{-3}$ in all our experiments), $d$ is the Euclidean distance and $d_m^2$ is a running average of the squared Euclidean distance of the $k$-th nearest neighbors. This running average is used to make the kernel more robust to the task being solved, as different tasks may have different typical distances between learnt embeddings. A detailed computation of the episodic reward can be found in Alg. 1 in App. A.1.

**Integrating life-long curiosity:** In principle, any long-term novelty estimator could be used as a basis for the modulator $\alpha_t$. We found Random Network Distillation (Burda et al., 2018b, RND) worked well, is simple to implement and easy to parallelize. The RND modulator $\alpha_t$ is defined by introducing a random, untrained convolutional network $g : \mathcal{O} \to \mathbb{R}^k$, and training a predictor network $\hat{g} : \mathcal{O} \to \mathbb{R}^k$ that attempts to predict the outputs of $g$ on all the observations that are seen during training by minimizing $\text{err}(x_t) = ||\hat{g}(x_t; \theta) - g(x_t)||^2$ with respect to the parameters of $\hat{g}$, $\theta$. We then define the modulator $\alpha_t$ as a normalized mean squared error, as done in Burda et al. (2018b): $\alpha_t = 1 + \frac{\text{err}(x_t) - \mu_e}{\sigma_e}$, where $\sigma_e$ and $\mu_e$ are running standard deviation and mean for $\text{err}(x_t)$. For more details about the architecture, see App. H.2, and hyperparameters, see App. F.

## 3 THE NEVER-GIVE-UP AGENT

In the previous section we described an episodic intrinsic reward for learning policies capable of maintaining exploration in a meaningful way throughout the agent's training process. We now demonstrate how to incorporate this intrinsic reward into a full agent that maintains a collection of value functions, each with a different exploration-exploitation trade-off.

Using intrinsic rewards as a means of exploration subtly changes the underlying Markov Decision Process (MDP) being solved: if the augmented reward $r_t = r_t^e + \beta r_t^i$ varies in ways unpredictable from the action and states, then the decision process may no longer be a MDP, but instead be a Partially Observed MDP (POMDP). Solving PODMPs can be much harder than solving MDPs, so to

avoid this complexity we take two approaches: firstly, the intrinsic reward is fed directly as an input to the agent, and secondly, our agent maintains an internal state representation that summarises its history of all inputs (state, action and rewards) within an episode. As the basis of our agent, we use Recurrent Replay Distributed DQN (Kapturowski et al., 2019, R2D2) as it combines a recurrent state, experience replay, off-policy value learning and distributed training, matching our desiderata.

Unlike most of the previously proposed intrinsic rewards (as seen in Section 1), the never-give-up intrinsic reward does not vanish over time, and thus the learned policy will always be partially driven by it. Furthermore, the proposed exploratory behaviour is directly encoded in the value function and as such it cannot be easily turned off. To overcome this problem, we proposed to jointly learn an explicit exploitative policy that is only driven by the extrinsic reward of the task at hand.

**Proposed architecture:** We propose to use a universal value function approximator (UVFA) $Q(x, a, \beta_i)$ to simultaneously approximate the optimal value function with respect to a family of augmented rewards of the form $r_t^{\beta_i} = r_t^e + \beta_i r_t^i$. We employ a discrete number $N$ of values $\{\beta_i\}_{i=0}^{N-1}$ including the special case of $\beta_0 = 0$ and $\beta_{N-1} = \beta$ where $\beta$ is the maximum value chosen. In this way, one can turn-off exploratory behaviour simply by acting greedily with respect to $Q(x, a, 0)$. Even before observing any extrinsic reward, we are able to learn a powerful representation and set of skills that can be quickly transferred to the exploitative policy. In principle, one could think of having an architecture with only two policies, one with $\beta_0 = 0$ and one with $\beta_1 > 0$. The advantage of learning a larger number of policies comes from the fact that exploitative and exploratory policies could be quite different from a behaviour standpoint. Having a larger number of policies that change smoothly allows for more efficient training. For a detailed description of the specific values of $\beta_i$ we use in our experiments, see App.A. We adapt the R2D2 agent that uses the dueling network architecture of Wang et al. (2015) with an LSTM layer after a convolutional neural network. We concatenate to the output of the network a one-hot vector encoding the value of $\beta_i$, the previous action $a_{t-1}$, the previous intrinsic reward $r_t^i$ and the previous extrinsic reward $r_t^e$. We describe the precise architecture in App. H.3 and hyperparameters in App. F.

**RL Loss functions:** As a training loss we use a transformed Retrace double Q-learning loss. In App. E we provide the details of the computation of the Retrace loss (Munos et al., 2016). In addition, we associate for each $\beta_i$ a $\gamma_i$, with $\gamma_0 = 0.997$, and $\gamma_{N-1} = 0.99$. We remark that the exploitative policy $\beta_0$ is associated with the highest discount factor $\gamma_0 = \gamma_{\max}$ and the most exploratory policy $\beta_{N-1}$ with the smallest discount factor $\gamma_0 = \gamma_{\min}$. We can use smaller discount factors for the exploratory policies because the intrinsic reward is dense and the range of values is small, whereas we would like the highest possible discount factor for the exploitative policy in order to be as close as possible from optimizing the undiscounted return. For a detailed description of the specific values of $\gamma_i$ we use in our experiments, see App. A.

**Distributed training:** Recent works in deep RL have achieved significantly improved performance by running on distributed training architectures that collect large amounts of experience from many actors running in parallel on separate environment instances (Andrychowicz et al., 2018; Barth-Maron et al., 2018; Burda et al., 2018b; Espeholt et al., 2018; Horgan et al., 2018; Kapturowski et al., 2019; Silver et al., 2016). Our agent builds upon the work by Kapturowski et al. (2019) to decouple learning from acting, with actors (256 unless stated otherwise) feeding experience into a distributed replay buffer and the learner training on randomly sampled batches from it in a prioritized way (Schaul et al., 2015b). Please refer to App. A for details.

## 4 EXPERIMENTS

We begin by analyzing the exploratory policy of the Never Give Up (NGU) agent with a single reward mixture. We perform such analysis by using a minimal example environment in Section 4.1. We observe the performance of its learned policy, as well as highlight the importance of learning a representation for abstract states. In Section 4.2, we analyze the performance of the full NGU agent, evaluating its effectiveness on the Arcade Learning Environment (ALE; Bellemare et al. (2013)). We measure the performance of the agent against baselines on hard exploration games, as well as dense reward games. We expand on the analysis of the NGU agent by running it on the full set of Atari games, as well as showing multiple ablations on important choices of hyperparameters of the model.

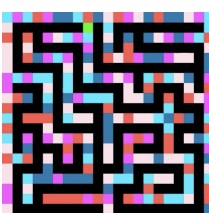 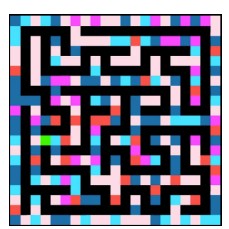 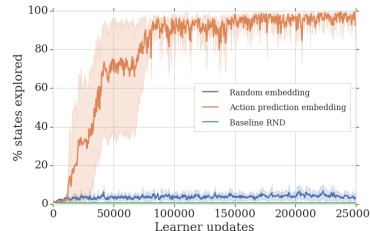

Figure 2: (Left and Center) Sample screens of Random Disco Maze. The agent is in green, and pathways in black. The colors of the wall change at every time step. (Right) Learning curves for Random projections vs. learned controllable states and a baseline RND implementation.

## 4.1 CONTROLLED SETTING ANALYSIS

In this section we present a simple example to highlight the effectiveness of the exploratory policy of the NGU agent, as well as the importance of estimating the exploration bonus using a controllable state representation. To isolate the effect of the exploratory policy, we restrict the analysis to the case of a single exploratory policy ($N = 1$, with $\beta = 0.3$). We introduce a gridworld environment, *Random Disco Maze*, implemented with the pycolab game engine (Stepleton, 2017), depicted in Fig. 2 (left). At each episode, the agent finds itself in a new randomly generated maze of size 21x21. The agent can take four actions {left, right, up, down}, moving a single position at a time. The environment is fully observable. If the agent steps into a wall, the episode ends and a new maze is generated. Crucially, at every time step, the color of each wall fragment is randomly sampled from a set of five possible colors, enormously increasing the number of possible states. This irrelevant variability in color presents a serious challenge to algorithms using exploration bonuses based on novelty, as the agent is likely to never see the same state twice. This experiment is purely exploratory, with no external reward. The goal is to see if the proposed model can learn a meaningful directed exploration policy despite the large visual distractions providing a continual stream of observation novelty to the agent. Fig. 2 shows the percentage of unique states (different positions in the maze) visited by agents trained with the proposed model and one in which the mapping $f$ is a fixed random projection (i.e. $f$ is untrained). The proposed model learns to explore any maze sampled from the task-distribution. The agent learns a strategy that resembles depth-first search: it explores as far as possible along each branch before backtracking (often requiring backtracking a few dozen steps to reach an unexplored area). The model with random projections, as well as our baseline of RND, do not show such exploratory behaviour[1]. Both models do learn to avoid walking into walls, doing so would limit the amount of intrinsic reward it would receive. However, staying alive is enough: simply oscillating between two states will produce different (and novel) controllable states at every time step.

## 4.2 ATARI RESULTS

In this section, we evaluate the effectiveness of the NGU agent on the Arcade Learning Environment (ALE; (Bellemare et al., 2013)). We use standard Atari evaluation protocol and pre-processing as described in Tab. 8 of App. F.4, with the only difference being that we do not use frame stacking. We restrict NGU to using the same setting and data consumption as R2D2, the best performing algorithm on Atari (Kapturowski et al., 2019). While we compare our results with the best published methods on this benchmark, we note that different baselines use very different training regimes with very different computational budgets. Comparing distributed and non-distributed methods is in general difficult. In an effort to properly assess the merits of the proposed model we include two additional baselines: as NGU is based on R2D2 using the Retrace loss (instead of its n-step objective) we include this as a baseline, and since we use RND as a reward modulator, we also include R2D2 with Retrace using the RND intrinsic reward. These methods are all run for 35 billion frames using the same protocol as that of R2D2 (Kapturowski et al., 2019). We detail the use of compute resources of the algorithms in App. D. We report the return averaged over 3 different seeds.

**Architecture:** We adopt the same core architecture as that used by the R2D2 agent to facilitate comparisons. There are still a few choices to make, namely: the size of the learned controllable states,

---

[1]See video of the trained agent here: `https://youtu.be/9HTY4ruPrHw`

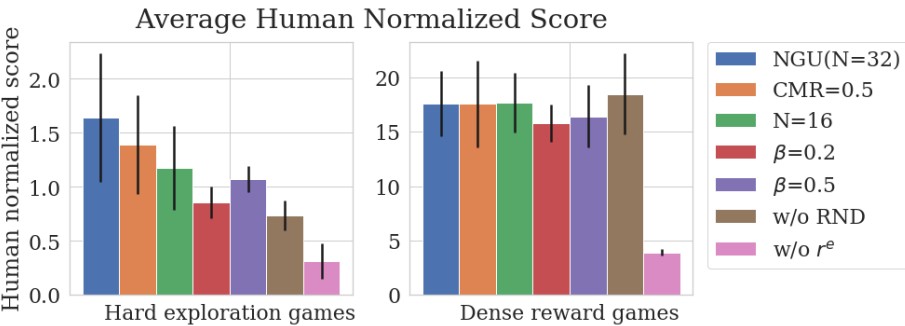

Figure 3: Human Normalized Scores on dense reward and hard exploration games.

the clipping factor $L$ in (1), and the number of nearest neighbours to use for computing pseudo-counts in (2). We selected these hyperparameters by analysing the performance of the single policy agent, NGU($N = 1$), on two representative exploration games: *Montezuma's Revenge* and *Pitfall!*. We report this study in App. B. We used the same fixed set of hyperparameters in all the remaining experiments.

**NGU agent:** We performed further ablations in order to better understand several major design choices of the full NGU agent on a set of 8 Atari games: the set of 5 dense reward games chosen to select the hyperparameters of Mnih et al. (2015), as well as 3 hard exploration games (*Montezuma's Revenge*, *Pitfall!*, and *Private Eye*). For a detailed description of the results on these games as well as results on more choices of hyperparameters, please see App.C. The ablations we perform are on the number of mixtures $N$, the impact of the off-policy data used (referred to as CMR below), the maximum magnitude of $\beta$ (by default $0.3$ if not explicitly mentioned), the use of RND to scale the intrinsic reward, and the performance of the agent in absence of extrinsic rewards. We denote by Cross Mixture Ratio (CMR) the proportion in the training batches of experience collected using different values of $\beta_i$ from the one being trained. A CMR of $0$ means training each policy only with data produced by the same $\beta_i$, while a CMR of $0.5$ means using equal amounts of data produced by $\beta_i$ and $\beta_{j \neq i}$. Our base agent NGU has a CMR of $0$.

The results are shown in Fig. 3. Several conclusions can be extracted from these results: Firstly, sharing experience from all the actors (with CMR of $0.5$) slightly harms overall average performance on hard exploration games. This suggests that the power of acting differently for different conditioning mixtures is mostly acquired through the shared weights of the model rather than shared data. Secondly, we observe an improvement, on average, from increasing the number of mixtures $N$ on hard exploration games. Thirdly, as one can observe in analyzing the value of $\beta$, the value of $\beta = 0.3$ is the best average performing value, whereas $\beta = 0.2$ and $\beta = 0.5$ make the average performance worse on those hard exploration games. These values indicate, in this case, the limits in which NGU is either not having highly enough exploratory variants ($\beta$ too low) or policies become too biased towards exploratory behavior ($\beta$ too high). Further, the use of the RND factor seems to be greatly beneficial on these hard exploration games. This matches the great success of existing literature, in which long-term intrinsic rewards appear to have a great impact (Bellemare et al., 2016; Ostrovski et al., 2017; Choi et al., 2018). Additionally, as outlined above, the motivation behind studying these variations on this set of 8 games is that those hyperparameters are of general effect, rather than specific to exploration. However, surprisingly, with the exception of the case of removing the extrinsic reward, they seem to have little effect on the dense reward games we analyze (with all error bars overlapping). This suggests that NGU and its hyperparameters are relatively robust: as extrinsic rewards become dense, intrinsic rewards (and their relative weight to the extrinsic rewards) naturally become less relevant. Finally, even without extrinsic reward $r^e$, we can still obtain average superhuman performance on the 5 dense reward games we evaluate, indicating that the exploration policy of NGU is an adequate high performing prior for this set of tasks. That confirms the findings of Burda et al. (2018a), where they showed that there is a high degree of alignment between the intrinsic curiosity objective and the hand-designed extrinsic rewards of many game environments. The heuristics of surviving and exploring what is controllable seem to be highly general and beneficial, as we have seen in the Disco Maze environment in Section 4.1, as well as on Atari.

| Algorithm | Gravitar | MR | Pitfall! | PrivateEye | Solaris | Venture |
|---|---|---|---|---|---|---|
| Human | 3.4k | 4.8k | 6.5k | 69.6k | **12.3k** | 1.2k |
| Best baseline | **15.7k** | **11.6k** | 0.0 | 11k | 5.5k | 2.0k |
| RND | 3.9k | 10.1k | -3 | 8.7k | 3.3k | 1.9k |
| R2D2+RND | 15.6k±0.6k | 10.4k±1.2k | -0.5±0.3 | 19.5k±3.5k | 4.3k±0.6k | **2.7k±0.0k** |
| R2D2(Retrace) | 13.3k±0.6k | 2.3k±0.4k | -3.5±1.2 | 32.5k±4.7k | 6.0k±1.1k | 2.0k±0.0k |
| NGU(N=1)-RND | 12.4k±0.8k | 3.0k±0.0k | **15.2k±9.4k** | 40.6k±0.0k | 5.7k±1.8k | 46.4±37.9 |
| NGU(N=1) | 11.0k±0.7k | 8.7k±1.2k | 9.4k±2.2k | 60.6k±16.3k | 5.9k±1.6k | 876.3±114.5 |
| NGU(N=32) | 14.1k±0.5k | 10.4k±1.6k | 8.4k±4.5k | **100.0k±0.4k** | 4.9k±0.3k | 1.7k±0.1k |

Table 1: Results against exploration algorithm baselines. Best baseline takes the best result among R2D2 (Kapturowski et al., 2019), DQN + PixelCNN (Ostrovski et al., 2017), DQN + CTS (Bellemare et al., 2016), RND (Burda et al., 2018b), and PPO + CoEx (Choi et al., 2018) for each game.

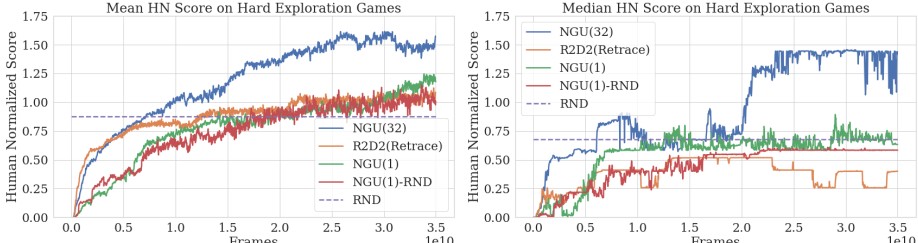

Figure 4: Human Normalized Scores on the 6 hard exploration games.

**Hard exploration games:** We now evaluate the full NGU agent on the six hard exploration games identified by Bellemare et al. (2016). We summarise the results on Tab. 1. The proposed method achieves on similar or higher average return than state-of-the-art baselines on all hard exploration tasks. Remarkably, to the best of our knowledge, this is the first method without use of privileged information that obtains a positive score on *Pitfall!*, with NGU($N = 1$)-RND obtaining a best score of 15,200. Moreover, in 4 of the 6 games, NGU($N = 32$) appears to substantially improve against the single mixture case NGU($N = 1$). This shows how the exploitative policy is able to leverage the shared weights with all the intrinsically-conditioned mixtures to explore games in which it is hard to do so, but still optimize towards maximizing the final episode score. In Fig. 4 we can see these conclusions more clearly: both in terms of mean and median human normalized scores, NGU greatly improves upon existing algorithms.

While direct comparison of the scores is interesting, the emphasis of this work is on learning directed exploration strategies that encourage the agent to cover as much of the environment as possible. In Fig. 4.2 (left) we observe the average episodic return of NGU run with and without RND on *Pitfall!*. NGU($N = 32$) is able to learn a directed exploration policy capable of exploring an average of 46 rooms per episode, crossing 14 rooms before receiving the first extrinsic reward. We also observe that, in this case, using RND makes our model be less data efficient. This is also the case for NGU($N = 1$), as observed on NGU($N = 1$)-RND in Tab. 1, the best performing *Pitfall!* agent. We conjecture three main hypotheses to explain this: firstly, on *Pitfall!* (and unlike *Montezuma's Revenge*) rooms are frequently aliased to one another, thus the agent does not obtain a large reward for discovering new rooms. This phenomenon would explain the results seen in Fig. 4.2 (right), in which RND greatly improves the results of NGU($N = 32$). Secondly, the presence of a timer in the observation acts as a spurious source of novelty which greatly increases the number of unique states achievable even within a single room. Thirdly, as analyzed in Section 3.7 of Burda et al. (2018b), RND-trained agents often keep 'interacting with danger' instead of exploring further, and *Pitfall!* is a game in which this can be highly detrimental, due to the high amount of dangerous elements in each room. Finally, we observe that NGU($N = 1$) obtains better results than NGU($N = 32$). Our intuition is that, in this case, a single policy should be simpler to learn and can achieve quite good results on this task, since exploration and exploitation policies are greatly similar.

**Dense reward games:** Tab. 2 shows the results of our method on dense reward games. NGU($N = 1$) underperforms relative to R2D2 on most games (indeed the same can be said of R2D2(Retrace) that serves as the basis of NGU). Since the intrinsic reward signal may be completely misaligned with the goal of the game, these results may be expected. However, there are cases such as Pong, in which NGU($N = 1$) catastrophically fails to learn to perform well. Here is where NGU($N = 32$) solves

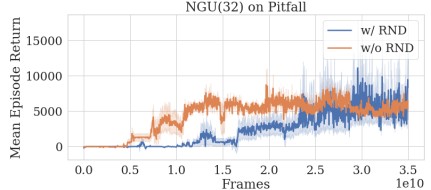 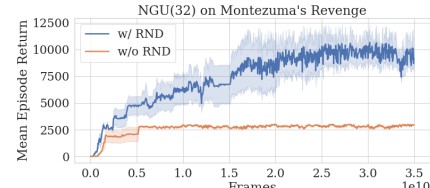

Figure 5: Mean episodic return for agents trained (left) *Pitfall!* and (right) *Montezuma's Revenge*.

| Algorithm | Pong | QBert | Breakout | Space Invaders | Beam Rider |
|---|---|---|---|---|---|
| Human | 14.6 | 13.4k | 30.5 | 1.6k | 16.9k |
| R2D2 | **21.0** | 408.8k | 837.7 | 43.2k | **188.2k** |
| R2D2+RND | 20.7±0.0 | 353.5k±41.0k | 815.8±5.3 | **54.5k±2.8k** | 85.7k±9.0k |
| R2D2(Retrace) | 20.9±0.0 | 415.6k±55.8k | 838.3±7.0 | 35.0k±13.0k | 111.1k±5.0k |
| NGU(N=1)-RND | -8.1±1.7 | 647.1k±50.5k | **864.0±0.0** | 45.3k±4.9k | 166.5k±8.6k |
| NGU(N=1) | -9.4±2.6 | **684.7k±8.8k** | **864.0±0.0** | 43.0k±3.9k | 114.6k±2.3k |
| NGU(N=32) | 19.6±0.1 | 465.8k±84.9k | 532.8±16.5 | 44.6k±1.2k | 68.7k±11.1k |

Table 2: Results against baselines on dense reward games.

this issue: the exploitative policy learned by the agent is able to reliably learn to play the game. Nevertheless, NGU($N = 32$) has limitations: even though its learned policies are vastly superhuman and empirically reasonable, they do not match R2D2 on Breakout and Beam Rider. This suggests that the representations learned by using the intrinsic signal still slightly interfere with the learning process of the exploitative mixture. We hypothesize that alleviating this further by having non-shared representations between mixtures should help in solving this issue.

**Results on all Atari 57 games:** The proposed method achieves an overall median score of 1354.4%, compared to 95% for Nature DQN baseline, 191.8% for IMPALA, 1920.6% for R2D2, and 1451.8% for R2D2 using retrace loss. Please refer to App. G for separate results on individual games. Even though its overall median score is lower than R2D2, NGU maintains good performance on all games, performing above human level on 51 out of the 57 games. This also shows further confirmation that the learned exploitative mixture is still able to focus on maximizing the score of the game, making the algorithm able to obtain great performance across all games.

**Analysis of Multiple Mixtures:** in Fig. 6, we can see NGU($N = 32$) evaluated with $\beta_0 = 0$ (used in all reported numerical results) against NGU($N = 32$) evaluated with $\beta_{31} = 0.3$. We can observe different trends in the games: on *Q\*Bert* the policies of the agent seem to converge to the exploitative policy regardless of the $\beta$ condition, with its learning curve being almost identical to the one shown for R2D2 in Kapturowski et al. (2019). As seen in App. G, this is common in many games. The second most common occurrence is what we see on *Pitfall!* and *Beam Rider*, in which the policies quantitatively learn very different behaviour. In these cases, the exploitative learns to focus on its objective, and sometimes it does so by benefiting from the learnings of the exploratory policy, as it is the case in *Pitfall!*[2], where R2D2 never achieves a positive score. Finally, there is the exceptional case of *Montezuma's Revenge*, in which the reverse happens: the exploratory policy obtains better score than the exploitative policy. In this case, extremely long-term credit assignment is required in order for the exploitative policy to consolidate the knowledge of the exploratory policy. This is because, to achieve scores that are higher than 16k, the agent needs to go to the second level of the game, going through many non-greedy and sometimes irreversible actions. For a more detailed analysis of this specific problem, see App. I.2.

## 5 CONCLUSIONS

We present a reinforcement learning agent that can effectively learn on both sparse and dense reward scenarios. The proposed agent achieves high scores in all Atari hard-exploration games, while still maintaining a very high average score over the whole Atari-57 suite. Remarkably, it is, to the best of our knowledge, the first algorithm to achieve non-zero rewards on the challenging game of *Pitfall!* without relying on human demonstrations, hand-crafted features, or manipulating the state of the environment. A central contribution of this work is a method for learning policies that can maintain

---

[2]See videos of NGU on Pitfall with $\beta_0$, $\beta_{31}$: https://sites.google.com/view/nguiclr2020

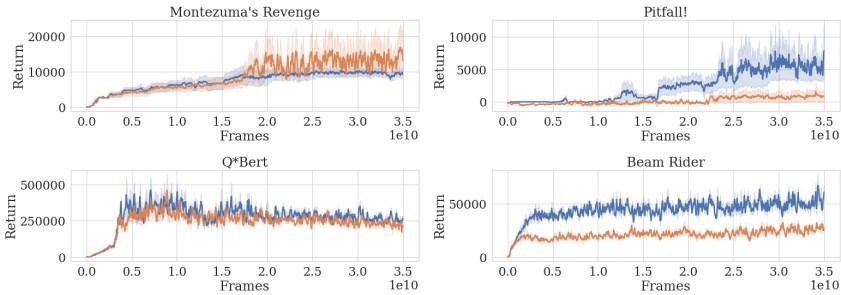

Figure 6: NGU(N=32) behavior for $\beta_0$ (blue) and $\beta_{31}$ (orange).

exploration throughout the training process. In the absence of extrinsic rewards, the method produces a policy that aims at traversing all controllable states of the MDP in a depth-first manner. We highlight that this could have impact beyond this specific application and/or algorithmic choices. For instance, one could use it as a behaviour policy to facilitate learning models of the environment or as a prior for planning methods.

The proposed method is able to leverage large amounts of compute by running on distributed training architectures that collect large amounts of experience from many actors running in parallel on separate environment instances. This has been crucial for solving most challenging tasks in deep RL in recent years (Andrychowicz et al., 2018; Espeholt et al., 2018; Silver et al., 2016), and this method is able to utilize such compute to obtain strong performance on the set of hard-exploration games on Atari. While this is certainly a desirable feature and allows NGU to achieve a remarkable performance, it comes at the price of high sample complexity, consuming a large amount of simulated experience taking several days of wall-clock time. An interesting avenue for future research lies in improving the data efficiency of these agents.

Further, the episodic novelty measure relies on the notion of controllable states to drive exploration. As observed on the Atari hard-exploration games, this strategy performs well on several tasks, but it may not be the right signal for some environments. For instance, in some environments it might take more than two consecutive steps to see the consequences of the actions taken by the agent. An interesting line for future research is learning effective controllable states beyond a simple inverse dynamics model.

Additionally, the proposed work relies on the assumption that while different, one can find good exploratory and exploitative policies that are similar enough to be effectively represented using a shared parameterization (implemented using the UVFA framework). This can be limiting when the two policies are almost adversarial. This can be seen in games such as 'Surround' and 'Ice hockey'.

Finally, the hyperparameter $\beta$ depends on the scale of the extrinsic reward. Thus, environments with significantly different extrinsic reward scales, might require different values of $\beta$. An interesting avenue forward is the dynamic adaptation of $\beta$, which could be done by using techniques such as Population Based Training (PBT)(Jaderberg et al., 2017) or Meta-gradients(Xu et al., 2018). Another advantage of dynamically tuning this hyperparameter would be to allow for the model to become completely exploitative when the agent has reached a point in which further exploring does not lead to improvements on the exploitative policy. This is not trivially achievable however, as including such a mechanism would require calibrating the adaptation to be aligned to the speed of learning of the exploitative policy.

## ACKNOWLEDGMENTS

We thank Daan Wierstra, Steph Hughes-Fitt, Andrea Banino, Meire Fortunato, Melissa Tan, Benigno Uria, Borja Ibarz, Mohammad Gheshlaghi Azar, Remi Munos, Bernardo Avila Pires, Andre Barreto, Vali Irimia, Sam Ritter, David Raposo, Tom Schaul and many other colleagues at DeepMind for helpful discussions and comments on the manuscript.

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

## A   EVALUATION SETUP

The evaluation we do is also identical to the one done in R2D2 Kapturowski et al. (2019): a parallel evaluation worker, which shares weights with actors and learners, runs the Q-network against the environment. This worker and all the actor workers are the two types of workers that draw samples from the environment. For Atari, we apply the standard DQN pre-processing, as used in R2D2. More concretely, this is how actors, evaluators, and learner are run:

**Learner**:

- Sample from the replay buffer a sequence of augmented rewards $r_t$, intrinsic rewards $r_t^i$, observations $x$, actions $a$ and discounts $\gamma_i$.
- Use Q-network to learn from $(r_t, x, a)$ with retrace using the procedure used by R2D2. As specified in Fig. 1, $r_t^i$ is sampled because it is fed as an input to the network.
- Use last 5 frames of the sampled sequences to train the action prediction network as specified in Section 2. This means that, for every batch of sequences, all time steps are used to train the RL loss, whereas only 5 time steps per sequence are used to optimize the action prediction loss.
- (If using RND) also use last 5 frames of the sampled sequences to train the predictor of RND as also specified in Section 2.

**Evaluator and Actor**

- Obtain $x_t$, $r_t^e$, $r_{t-1}^i$, and discount $\gamma_i$.
- With these inputs, compute forward pass of R2D2 to obtain $a_t$.
- With $x_t$, compute $r_t^i$ using the embedding network as described in Section 2.
- (actor) Insert $x_t$, $a_t$, $r_t = r_t^e + \beta_i r_t^i$, $\gamma_i$, and $r_t^i$ in the replay buffer.
- Step on the environment with $a_t$.

**Distributed training**

As in R2D2, we train the agent with a single GPU-based learner, performing approximately 5 network updates per second (each update on a mini-batch of 64 length-80 sequences, as explained below, and each actor performing $\sim 260$ environment steps per second on Atari. We assign to each actor a fixed value in the set $\{\beta_i\}_{i=0}^{N-1}$ and the actor acts according to an $\epsilon$-greedy version of this policy. More concretely for the $j$-th actor we assign the value $\beta_h$ with $h = j \mod N - 1$. In our experiments, we use the following $\beta_i$:

$$\beta_i = \begin{cases} 0 & \text{if } i = 0 \\ \beta & \text{if } i = N - 1 \\ \beta \cdot \sigma(10\frac{2i - (N-2)}{N-2}) & otherwise \end{cases}$$

where $\sigma$ is the sigmoid function. This choice of $\beta_i$, as you can see in Fig.7(a), allows to focus more on the two extreme cases which are the fully exploitative policy and very exploratory policy.

In the replay buffer, we store fixed-length sequences of $(x, a, r)$ tuples. In all our experiments we collect sequences of length 80 timesteps, where adjacent overlap by 40 time-steps. These sequences never cross episode boundaries. Additionally, we store in the replay the value of the $\beta_i$ used by the actor as well as the initial recurrent state, that we use to initialize the network at training time. Please refer to Kapturowski et al. (2019) for a detailed experimental of trade-offs on different treatments of recurrent states in replay. Given a single batch of trajectories we unroll both online and target networks on the same sequence of states to generate value estimates. We use prioritized experience replay. We followed the same prioritization scheme proposed in Kapturowski et al. (2019) using a mixture of max and mean of the TD-errors with priority exponent $\eta = 1.0$. In addition, we associate for each $\beta_i$ a $\gamma_i$ such that:

$$\gamma_i = 1 - \exp\left(\frac{(N - 1 - i)\log(1 - \gamma_{\max}) + i\log(1 - \gamma_{\min})}{N - 1}\right), \tag{4}$$

where $\gamma_{\max}$ is the maximum discount factor and $\gamma_{\min}$ is the minimal discount factor. This form allows to have discount factors evenly spaced in log-space between $1 - \gamma_{\max}$ and $1 - \gamma_{\min}$. For more intuition, we provide a graph of the $\{\gamma_i\}_{i=0}^{N-1}$ in Fig.7(b) in App.A. We remark that the exploitative policy $\beta_0$ is associated with the highest discount factor $\gamma_0 = \gamma_{\max}$ and the most exploratory policy $\beta_{N-1}$ with the smallest discount factor $\gamma_0 = \gamma_{\min}$. We can use smaller discount factors for the exploratory policies because the intrinsic reward is dense and the range of values is small, whereas we would like the highest possible discount factor for the exploitative policy in order to be as close as possible to optimizing the undiscounted return. In our experiments, we use $\gamma_{\max} = 0.997$ and $\gamma_{\min} = 0.99$.

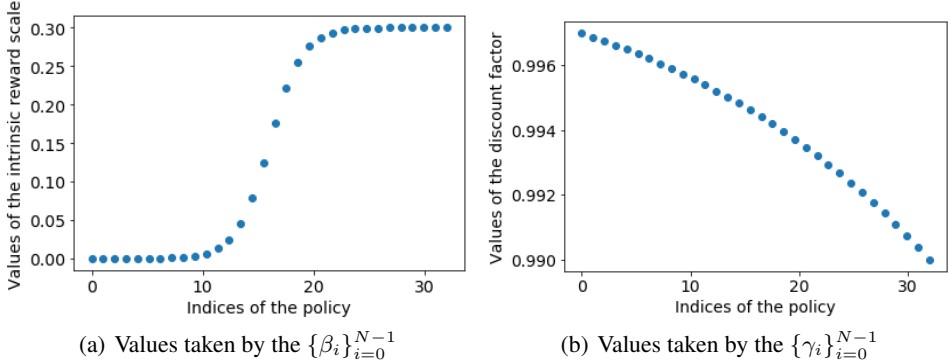

(a) Values taken by the $\{\beta_i\}_{i=0}^{N-1}$       (b) Values taken by the $\{\gamma_i\}_{i=0}^{N-1}$

Figure 7: Values taken by the $\{\beta_i\}_{i=0}^{N-1}$ and the $\{\gamma_i\}_{i=0}^{N-1}$ for $N = 32$ and $\beta = 0.3$.

## A.1 Never-give-up intrinsic reward algorithm

We present the algorithm for computing the intrinsic reward in Alg. 1. We follow the notations defined in Sec. 2 in the paragraph relative to the episodic intrinsic reward:

- $M$ the episodic memory containing at time $t$ the previous embeddings $\{f(x_0), f(x_1), \ldots, f(x_{t-1})\}$.
- $k$ is the number of nearest neighbours.
- $N_k = \{f_i\}_{i=1}^{k}$ is the set of $k$-nearest neighbours of $f(x_t)$ in the memory $M$.
- $K$ the kernel defined as $K(x, y) = \frac{\epsilon}{\frac{d^2(x,y)}{d_m^2} + \epsilon}$ where $\epsilon$ is a small constant, $d$ is the Euclidean distance and $d_m^2$ is a running average of the squared Euclidean distance of the $k$-nearest neighbors.
- $c$ is the pseudo-counts constant.
- $\xi$ cluster distance.
- $s_m$ maximum similarity.

## A.2 Complexity analysis

The space complexity is constant. The number of weights that the network has can be computed from the architecture seen in App. F. Furthermore, for our episodic memory buffer, we pre-allocate memory at the beginning of training, with size detailed in App. F. In cases in which the episode is longer than the size of the memory, the memory acts a ring buffer, deleting oldest entries first.

Time complexity is $O(M \cdot N)$, where $N$ is the number of frames, and $M$ is the size of our memory. This is due to the fact that we do one forward pass per frame, and we compute the distance from the embeddings produced by the embeddings network to the contents of our memory in order to retrieve the $k$-nearest neighbors.

---

**Algorithm 1:** Computation of the episodic intrinsic reward at time $t$: $r_t^{\text{episodic}}$.

---

**Input** : $M$; $k$; $f(x_t)$; $c$; $\epsilon$; $\xi$; $s_m$; $d_m^2$
**Output** : $r_t^{\text{episodic}}$

1 Compute the $k$-nearest neighbours of $f(x_t)$ in $M$ and store them in a list $N_k$
2 Create a list of floats $d_k$ of size $k$
```
/* The list d_k will contain the distances between the embedding
   f(x_t) and its neighbours N_k.                                      */
```
3 **for** $i \in \{1, \ldots, k\}$ **do**
4 $\quad$ $d_k[i] \leftarrow d^2(f(x_t), N_k[i])$
5 **end**
6 Update the moving average $d_m^2$ with the list of distances $d_k$
```
/* Normalize the distances d_k with the updated moving average d_m^2.
                                                                        */
```
7 $d_n \leftarrow \frac{d_k}{d_m^2}$
```
/* Cluster the normalized distances d_n i.e.  they become 0 if too
   small and 0_k is a list of k zeros.                                 */
```
8 $d_n \leftarrow \max(d_n - \xi, 0_k)$
```
/* Compute the Kernel values between the embedding f(x_t) and its
   neighbours N_k.                                                     */
```
9 $K_v \leftarrow \frac{\epsilon}{d_n + \epsilon}$
```
/* Compute the similarity between the embedding f(x_t) and its
   neighbours N_k.                                                     */
```
10 $s \leftarrow \sqrt{\sum_{i=1}^{k} K_v[i]} + c$
```
/* Compute the episodic intrinsic reward at time t:  r_t^i.           */
```
11 **if** $s > s_m$ **then**
12 $\quad$ $r_t^{\text{episodic}} \leftarrow 0$
13 **else**
14 $\quad$ $r_t^{\text{episodic}} \leftarrow \frac{1}{s}$

---

# B  ABLATIONS FOR NGU(N=1)

As mentioned in Section 4.2, we here show ablations on the size of the learned controllable states, the clipping factor $L$ in (1), and the number of nearest neighbours to use for computing pseudo-counts in (2).

Due to the lack of a pure exploitative mode, as seen in 4.2, NGU(N=1) fails to perform well in dense reward games. Therefore, in order to obtain high signal from these ablations, we analyze the performance of NGU(N=1) on the two most popular sparse reward games: *Montezuma's Revenge* and *Pitfall!*.

## B.1  SIZE OF CONTROLLABLE STATES

In Fig. 8 and Fig. 9 we can see the performance of NGU(N=1) with different sizes of the size of the controllable state on *Pitfall!* and *Montezuma's Revenge* respectively. As we can observe, that there is small to no impact on *Pitfall!*, with scores that sometimes reach more than 25,000 points. On *Montezuma's Revenge* 32 is the value that is consistently better than 64. A size of 16 as the controllable state size sometimes solves the level, but is in general less stable.

## B.2  NEAREST NEIGHBORS USED

We proceed to show a similar analysis on Fig. 10 and Fig. 11 regarding the amount of nearest neighbors on *Pitfall!* and *Montezuma's Revenge* respectively. As we can see, there are slight gains

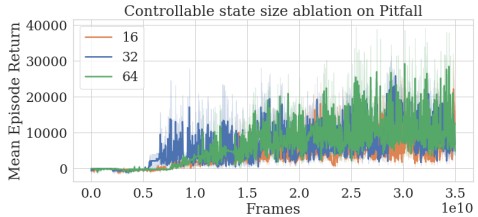

Figure 8: Mean episodic return for agents trained *Pitfall!*.

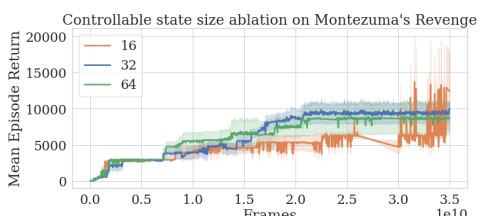

Figure 9: Mean episodic return for agents trained *Montezuma's Revenge*.

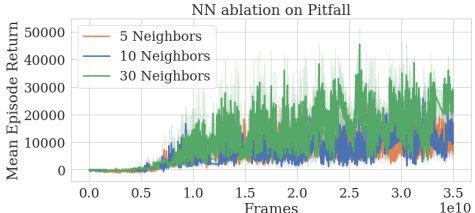

Figure 10: Mean episodic return for agents trained *Pitfall!*.

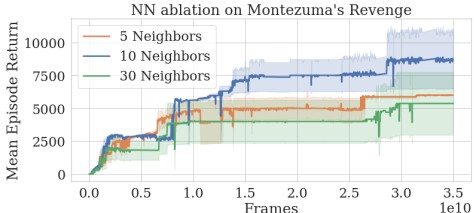

Figure 11: Mean episodic return for agents trained *Montezuma's Revenge*.

from using more neighbors on *Pitfall!*, whereas there is a clear difference in performance in using 10 neighbors in *Montezuma's Revenge* when compared to using 5 or 30 neighbors.

### B.3 CLIPPING FACTOR L

Finally, we show the performance of NGU(N=1) on Fig. 12 and Fig. 13 regarding the clipping factor $L$ *Pitfall!* and *Montezuma's Revenge* respectively. As we can observe, *Pitfall!* is again robust to the value of this hyperparameter, with marginally worse performance in the case of $L = 10$. This is expected, as RND is generally detrimental to the performance of NGU on *Pitfall!*, as seen in Section 4.2. On the other hand, the highest value of clipping appears to work best on *Montezuma's Revenge*. In our initial investigations, we observed that clipping this value was required on *Montezuma's Revenge* to make the algorithm stable. Further analysis is required in order to show the range of values of $L$ that are higher than 10 and are detrimental to the performance of NGU(N=1) on this task.

## C ABLATIONS FOR NGU(N=32)

### C.1 GENERAL ABLATIONS

Tab. 3 shows the results for all the ablations we performed on 8 games for NGU($N = 32$). We can see that the conclusions of Sec. 4.2 hold, with a few additional facts to observe:

- The best score in *Montezuma's Revenge* is obtained by using a non-zero Cross Mixture Ratio, even though it is relatively close to the score obtained by NGU($N = 32$).

- $N = 2$ and $N = 8$ have lower average human normalized score on the set of 3 hard exploration games when compared to $N = 16$ or $N = 32$. Concretely on the set of hard

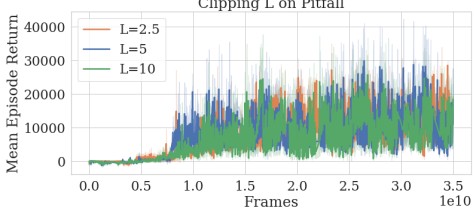

Figure 12: Mean episodic return for agents trained *Pitfall!*.

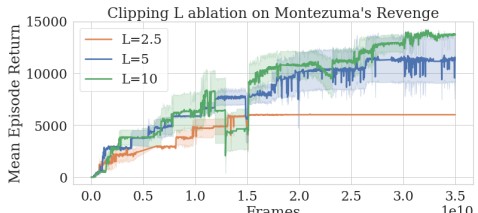

Figure 13: Mean episodic return for agents trained *Montezuma's Revenge*.

| Algorithm | Pong | Qbert | Breakout | Space Invaders | Beam Rider | MR | Pitfall! | PrivateEye |
|-----------|------|-------|----------|----------------|------------|-----|----------|------------|
| Human | 14.6 | 13.4k | 30.5 | 1.6k | 16.9k | 4.8k | 6.5k | 69.6k |
| NGU(N=32) | 19.6±0.1 | 465.8k±84.9k | 532.8±16.5 | 44.6k±1.2k | 68.7k±11.1k | 10.4k±1.6k | **8.4k±4.5k** | **100.0k±0.4k** |
| N=2 | **20.6±0.1** | 457.7k±128.0k | **576.0±24.9** | **48.0k±6.2k** | 71.8k±9.0k | 11.1k±1.4k | -1.6±0.9 | 40.6k±0.1k |
| N=8 | 20.0±0.3 | 481.8k±41.2k | 524.1±28.3 | 43.1k±5.5k | 64.0k±3.8k | 7.8k±0.3k | -1.9±0.4 | 38.5k±1.6k |
| N=16 | 17.2±1.0 | 444.2k±61.7k | 549.0±11.0 | 46.4k±3.5k | 73.9k±5.3k | 9.5k±0.8k | 5.0k±1.2k | 52.0k±20.7k |
| CMR=0.5 | 19.0±0.3 | 502.9k±73.9k | 516.9±20.1 | 40.3k±7.1k | 74.5k±5.9k | **12.0k±0.8k** | 5.1k±2.4k | 58.8k±16.9k |
| β=0.2 | 20.3±0.2 | 350.1k±24.7k | 525.0±38.2 | 43.9k±1.3k | 75.6k±10.9k | 6.9k±0.1k | 3.3k±1.4k | 40.6k±0.0k |
| β=0.5 | 16.8±1.2 | 480.4k±59.2k | 451.6±17.2 | 40.2k±4.8k | 62.3k±5.9k | 11.3k±0.5k | 1.3k±0.5k | 44.5k±3.2k |
| w/o RND | 19.7±0.4 | **550.3k±64.4k** | 553.7±19.1 | 47.6k±3.7k | **87.1k±9.1k** | 3.0k±0.0k | 7.7k±0.9k | 40.5k±0.1k |
| w/o $r^e$ | -8.4±1.9 | 28.1k±1.1k | 383.0±10.0 | 5.5k±0.3k | 6.4k±0.2k | 3.4k±0.7k | 600.4±468.9 | 7.5k±2.2k |

Table 3: Ablation results on 8 Atari games.

| Algorithm | Gravitar | Solaris | Venture |
|-----------|----------|---------|---------|
| Human | 3.4k | 12.3k | 1.2k |
| NGU(N=32) | 14.1k±0.5k | 4.9k±0.3k | 1.7k±0.1k |
| β=0.2 | 14.2k±0.3k | 6.4k±1.4k | 2.3k±0.2k |
| β=0.5 | 13.5k±0.4k | 5.5k±1.4k | 2.0k±0.1k |

Table 4: Further ablation results on hard exploration games.

exploration games of Tab. 3, they only achieve super-human performance on *Montezuma's Revenge*.

- Even though we have seen that the results of $\beta = 0.2$ and $\beta = 0.5$ have lower average on the 3 hard exploration games of Tab. 3, they still individually outperform RND, R2D2, R2D2(Retrace), and R2D2+RND on *Pitfall!* and *Private Eye*.

- In the case of *Private Eye* the distance in score might be misleading, as rewards are very sparse of large value. For instance, after reaching a score of 40k, if we ignore minor rewards, there are only two rewards to be collected of around 30k points. This creates what seems to be large differences in scores.

- On Breakout, a high score is achieved without extrinsic reward. This is due to the fact that the exploratory policy learns to survive, which eventually leads to a high score.

## C.2 FURTHER ABLATIONS ON HARD EXPLORATION

On Tab.4 we show further results on the case of $\beta = 0.2$ and $\beta = 0.5$. We compare them to human performance as well as the base NGU($N = 32$), with $\beta = 0.3$.

As we can observe, in this case the difference in terms of relative performance among games is less pronounced than the ones observed on Tab. 3. In fact, results are slightly better for both values of $\beta$ on all 3 games, with a maximum difference of 1.5k points on Solaris between $\beta = 0.3$ and $\beta = 0.2$. We hypothesize that this is due to the nature of these specific games: the policies learnt on these three games seem to focus on exploitation rather than extended exploration of the environment, and in that case, similar to what we see for dense reward games in Sec. 4.2, the method shows less variability with respect to this hyperparameter.

## D ALGORITHM COMPUTATION COMPARISON

On Tab. 5 we can see a comparison of the computation used between different algorithms.

Computation is still difficult to compare even when taking actor steps and parameter updates into account: distributed the number of actors in distributed setups will affect how much data the learner will be able to consume, but also how off-policy such data is (e.g. in R2D2, if a learner is learning from many actors, the data that is sampled from the replay buffer will be more recent than with fewer actors).

## E DETAILS ON THE RETRACE ALGORITHM

Retrace (Munos et al., 2016) is an off-policy Reinforcement Learning algorithm that can be used for evaluation or control. In the evaluation setting the goal is mainly to estimate the action-value function $Q^\pi$ of a target policy $\pi$ from trajectories drawn from a behaviour policy $\mu$. In the control

| Algorithm | Number of actors | Total Number of frames |
|---|---|---|
| R2D2 Kapturowski et al. (2019) | 256 | 35 B |
| R2D2(Retrace) | 256 | 35 B |
| R2D2 + RND | 256 | 35 B |
| NGU-RND(N=1) | 256 | 35 B |
| NGU (N=1) | 256 | 35 B |
| NGU (N=32) | 256 | 35 B |
| DQN + PixelCNN Ostrovski et al. (2017) | 1 | 150 M |
| DQN + CTS Bellemare et al. (2016) | 1 | 150 M |
| RND** Burda et al. (2018b) | 1024 | 16 B (2 B) |
| PPO + CoEx Choi et al. (2018) | 1024 | 2 B |

Table 5: Comparison in number of steps.
**To obtain its best reported result on Montezuma's Revenge 16B (which we compare against), the results table of the work are with 2B.

setting the target policy, or more precisely the sequence of target policies, depends on the sequence of $Q$-functions that will be generated through the process of approximating $Q^*$. To do so, we consider trajectories $\tau$ starting from the state-action couple $(x, a)$ and then following the behaviour policy $\mu$ of the form:

$$\tau = (x_t, a_t, r_t, x_{t+1})_{t \in \mathbb{N}}, \tag{5}$$

with $(x_0, a_0) = (x, a)$, $\forall t \geq 1, a_t \sim \mu(.|x_t)$, $\forall t \geq 0, r_t = r(x_t, a_t)$ and $\forall t \geq 0, x_{t+1} \sim P(.|x_t, a_t)$. The expectation $\mathbb{E}_\mu$ is over all admissible trajectories $\tau$ generated by the behaviour policy $\mu$ starting in state $x$ doing action $a$ and then following the behaviour policy $\mu$.

The general Retrace operator $\mathcal{T}$, that depends on $\mu$ and $\pi$, is:

$$\mathcal{T}Q(x, a) = Q(x, a) + \mathbb{E}_\mu \left[ \sum_{t \geq 0} \gamma^t \left( \prod_{s=1}^{t} c_s \right) \delta_t \right], \tag{6}$$

where the temporal difference $\delta_t$ is defined as:

$$\delta_t = r_t + \gamma \sum_{a \in A} \pi(a|x_{t+1}) Q(x_{t+1}, a) - Q(x_t, a_t), \tag{7}$$

and the cutting traces coefficients $c_s$ as:

$$c_s = \lambda \min \left( 1, \frac{\pi(a_s|x_s)}{\mu(a_s|x_s)} \right). \tag{8}$$

Theorem 2 of Munos et al. (2016) explains in which conditions the sequence of $Q$-functions:

$$Q_{k+1} = \mathcal{T}_k Q_k, \tag{9}$$

where $\mathcal{T}_k$ depends on the policy-couple $(\mu_k, \pi_k)$ converges to the optimal $Q$-value $Q^*$. In particular one of the conditions is that the sequence of target policies $\pi_k$ is greedy or $\epsilon$-greedy with respect to $Q_k$ (more details can be found in Munos et al. (2016)).

In practice, at a given time $t$, we can only consider finite sampled sequences $(x_s, a_s, r_s, x_{s+1})_{s=t}^{t+k}$ starting from $(x_t, a_t)$ and then following the behaviour policy $\mu$. Therefore, we define the finite sampled-Retrace operator as:

$$\hat{T}Q(x_t, a_t) = Q(x_t, a_t) + \sum_{s=t}^{t+k-1} \gamma^{s-t} \left( \prod_{i=t+1}^{s} c_i \right) \delta_s. \tag{10}$$

In addition, we use two neural networks. One target network $Q(x, a; \theta^-)$ and an online network $Q(x, a; \theta)$. The target network is used to compute the target value $\hat{y}_t$ that the online network will try to fit:

$$\hat{y}_t = \hat{T}Q(x_t, a_t; \theta^-), \tag{11}$$

$$= Q(x_t, a_t; \theta^-) + \sum_{s=t}^{t+k-1} \gamma^{s-t} \left( \prod_{i=t+1}^{s} c_i \right) \left( r_s + \gamma \sum_{a \in A} \pi(a|x_{s+1}) Q(x_{s+1}, a; \theta^-) - Q(x_s, a_s; \theta^-) \right). \tag{12}$$

In the control scenario the policy chosen $\pi(a|x)$ is greedy or $\epsilon$-greedy with respect to the online network $Q(x, a; \theta)$. Then, the online network is optimized to minimize the loss:

$$L(x_t, a_t, \theta) = (Q(x_t, a_t; \theta) - \hat{y}_t)^2. \tag{13}$$

More generally, one can use transformed Retrace operators(Pohlen et al., 2018):

$$\mathcal{T}^h Q(x, a) = \mathbb{E}_\mu \left[ h \left( h^{-1}(Q(x, a)) + \sum_{t \geq 0} \gamma^t \left( \prod_{s=1}^{t} c_s \right) \delta_t^h \right) \right], \tag{14}$$

where $h \in \mathbb{R}^\mathbb{R}$ is a real-function and the temporal difference $\delta_t^h$ is defined as:

$$\delta_t^h = r_t + \gamma \sum_{a \in A} \pi(a|x_{t+1}) h^{-1}(Q(x_{t+1}, a)) - h^{-1}(Q(x_t, a_t)). \tag{15}$$

The role of the function $h$ is to reduce (squash) the scale of the action-value function to make it easier to approximate for a neural network without changing the optimal property of the operator $\mathcal{T}$. In particular, we use the function $h$:

$$\forall z \in \mathbb{R}, h(z) = \text{sign}(z)(\sqrt{|z| + 1} - 1) + \epsilon z, \tag{16}$$

$$\forall z \in \mathbb{R}, h^{-1}(z) = \text{sign}(z) \left( \left( \frac{\sqrt{1 + 4\epsilon(|z| + 1 + \epsilon)} - 1}{2\epsilon} \right) - 1 \right), \tag{17}$$

with $\epsilon = 10^{-2}$.

# F HYPERPARAMETERS

## F.1 SELECTION OF HYPERPARAMETERS

In order to select the hyperparameters used for NGU($N = 32$) for all 57 Atari games, which are shown on Tab. 6, we ran a grid search with the ranges shown on Tab. 9. We used 3 seeds on the set of 8 Atari games shown in Tab. 3. Regarding the hyperparameters concerning the kernel $K$ (Kernel $\epsilon$ and the number of neighbors used), we fixed them after determining suitable ranges of the intrinsic reward in our initial experimentation on Atari. After running the grid search with those hyperparameters, we selected the combination with the highest amount games (out of 8) that held a score greater than our human benchmark. As one can see on the multiple mixtures ablations seen on Tab. 3, as well as the single mixture ablations on App B, the only agent that achieved superhuman performance on the set of 8 games is NGU($N = 32$).

Finally, in order to obtain the R2D2+RND baseline, we ran a sweep over the $\beta$ hyperparameter with values 0.1, 0.3, and 0.5, over the 8 games shown in Tab. 3. Coincidentally, like NGU($N = 32$), the best value of $\beta$ was determined to be 0.3.

## F.2 COMMON HYPERPARAMETERS

These are the hyperparameters used in all the experiments. We expose a full list of hyperparameters here for completeness. However, as one can see, the R2D2-related architectural hyperparameters are identical to the original R2D2 hyperparameters. Shown in Tab. 6.

| Hyperparameter | Value |
|---|---|
| Number of Seeds | 3 |
| Cross Mixture Ratio | 0.0 |
| Number of mixtures $N$ | 32 |
| Optimizer | AdamOptimizer (for all losses) |
| Learning rate (R2D2) | 0.0001 |
| Learning rate (RND and Action prediction) | 0.0005 |
| Adam epsilon | 0.0001 |
| Adam beta1 | 0.9 |
| Adam beta2 | 0.999 |
| Adam clip norm | 40 |
| Discount $r^i$ | 0.99 |
| Discount $r^e$ | 0.997 |
| Batch size | 64 |
| Trace length | 80 |
| Replay period | 40 |
| Retrace $\lambda$ | 0.95 |
| R2D2 reward transformation | $\text{sign}(x) \cdot (\sqrt{|x| + 1} - 1) + 0.001 \cdot x$ |
| Episodic memory capacity | 30000 |
| Embeddings memory mode | Ring buffer |
| Intrinsic reward scale $\beta$ | 0.3 |
| Kernel $\epsilon$ | 0.0001 |
| Kernel num. neighbors used | 10 |
| Kernel cluster distance $\xi$ | 0.008 |
| Kernel pseudo-counts constant $c$ | 0.001 |
| Kernel maximum similarity $s_m$ | 8 |
| Replay priority exponent | 0.9 |
| Replay capacity | $5e6$ |
| Minimum sequences to start replay | 6250 |
| Actor update period | 100 |
| Target Q-network update period | 1500 |
| Embeddings target update period | once/episode |
| Action prediction network L2 weight | 0.00001 |
| RND clipping factor $L$ | 5 |
| Evaluation $\epsilon$ | 0.01 |
| Target $\epsilon$ | 0.01 |

Table 6: Common hyperparameters.

## F.3 DISCO MAZE HYPERPARAMETERS

Hyperparameters are shown in Tab. 7.

| Hyperparameter | Value |
|---|---|
| Episodic memory capacity | 5000 |
| Learning rate (R2D2 and Action prediction) | 0.001 |
| Replay capacity | $1e6$ |
| Intrinsic reward scale $\beta$ | 0.5 |
| Trace length | 50 |
| Replay period | 50 |
| Retrace $\lambda$ | 0.97 |
| Retrace loss transformation | $identity$ |
| Num. action repeats | 1 |
| Target Q-network update period | 100 |
| Q-network filter sizes | $(3, 3)$ |
| Q-network filter strides | $(1, 1)$ |
| Q-network num. filters | $(16, 32)$ |
| Action prediction network filter sizes | $(3, 3)$ |
| Action prediction network filter strides | $(1, 1)$ |
| Action prediction network num. filters | $(16, 32)$ |
| Kernel $\epsilon$ | 0.01 |
| Evaluation $\epsilon$ | 0 |

Table 7: Disco Maze hyperparameters.

## F.4 ATARI PRE-PROCESSING HYPERPARAMETERS

Hyperparameters are shown in Tab. 8.

| Hyperparameter | Value |
|---|---|
| Max episode length | $30\ min$ |
| Num. action repeats | 4 |
| Num. stacked frames | 1 |
| Zero discount on life loss | $false$ |
| Random noops range | 30 |
| Sticky actions | $false$ |
| Frames max pooled | 3 and 4 |
| Grayscaled/RGB | Grayscaled |
| Action set | Full |

Table 8: Atari pre-processing hyperparameters.

## F.5 HYPERPARAMETER RANGES

On Tab. 9 we can see the ranges we used to sweep over in our experiments.

| Hyperparameter | Value |
|---|---|
| Intrinsic reward scale $\beta$ | $\{0.2,\ 0.3,\ 0.5\}$ |
| Number of mixtures $N$ | $\{1,\ 2,\ 8,\ 16,\ 32\}$ |
| Cross Mixture Ratio | $\{0.0,\ 0.25,\ 0.5\}$ |
| # Episodes w/o wiping Episodic Memory | $\{1,\ 3\}$ |

Table 9: Range of hyperparameters sweeps.

# G    DETAILED ATARI RESULTS

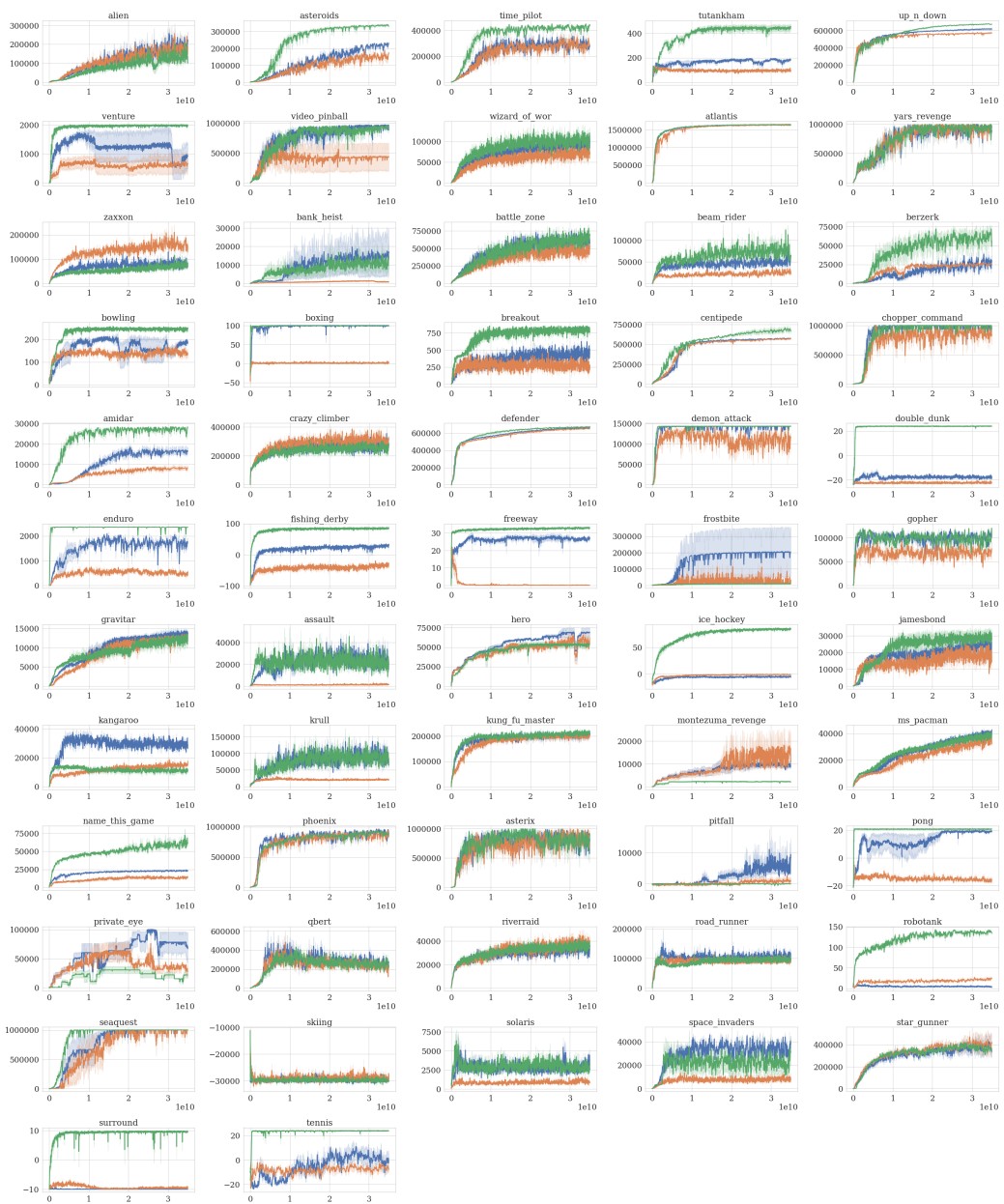

Figure 14: R2D2(Retrace) (green), NGU(N=32) with eval $\beta = 0.0$ (blue) and eval $\beta = 0.3$ (orange).

| Game | R2D2(Retrace) | NGU(32) eval beta=0.0 | NGU(32) eval beta=0.3 |
|---|---|---|---|
| alien | 189.1k±15.6k | **248.1k±22.4k** | 225.5k±36.9k |
| asteroids | **338.5k±4.5k** | 230.5k±4.0k | 181.6k±2.8k |
| time pilot | **446.8k±3.9k** | 344.7k±31.0k | 336.1k±32.9k |
| tutankham | **452.2±20.0** | 191.1±1.2 | 125.7±5.2 |
| up n down | **678.8k±1.3k** | 620.1k±13.7k | 575.2k±10.4k |
| venture | **2.0k±0.0k** | 1.7k±0.1k | 779.9±188.7 |
| video pinball | 948.6k±5.2k | **965.3k±12.8k** | 596.7k±120.0k |
| wizard of wor | **120.2k±7.6k** | 106.2k±7.0k | 85.1k±12.3k |
| atlantis | **1654.9k±1.5k** | 1653.6k±2.3k | 1638.0k±4.6k |
| yars revenge | 990.4k±2.9k | 986.0k±3.2k | **993.8k±1.5k** |
| zaxxon | 94.8k±23.7k | 111.1k±11.8k | **190.1k±6.9k** |
| bank heist | 15.0k±5.7k | **17.4k±12.0k** | 1.4k±0.0k |
| battle zone | **733.1k±45.2k** | 691.7k±22.6k | 571.5k±45.4k |
| beam rider | **103.9k±1.1k** | 63.6k±8.6k | 31.9k±0.6k |
| berzerk | **69.3k±5.8k** | 36.2k±4.7k | 27.3k±0.8k |
| bowling | **253.0±1.2** | 211.9±8.4 | 161.8±4.9 |
| boxing | **100.0±0.0** | 99.7±0.0 | 3.8±1.4 |
| breakout | **839.7±6.0** | 559.2±29.0 | 387.6±51.0 |
| centipede | **700.2k±19.8k** | 577.8k±3.1k | 574.6k±7.0k |
| chopper command | **999.9k±0.0k** | 999.9k±0.0k | 974.1k±14.7k |
| amidar | **28.2k±0.2k** | 17.8k±1.2k | 8.5k±0.4k |
| crazy climber | 294.2k±25.7k | 313.4k±7.7k | **357.0k±8.3k** |
| defender | **675.6k±3.2k** | 664.1k±1.8k | 656.3k±7.2k |
| demon attack | **143.9k±0.0k** | 143.5k±0.0k | 136.6k±3.3k |
| double dunk | **24.0±0.0** | -14.1±2.4 | -21.5±0.7 |
| enduro | **2.4k±0.0k** | 2.0k±0.1k | 682.5±24.0 |
| fishing derby | **86.8±0.9** | 32.0±1.9 | -29.3±4.1 |
| freeway | **33.2±0.1** | 28.5±1.1 | 21.2±0.5 |
| frostbite | 10.5k±3.8k | **206.4k±150.0k** | 67.9k±45.6k |
| gopher | **117.8k±0.9k** | 113.4k±0.6k | 86.6k±7.4k |
| gravitar | 12.9k±0.6k | **14.2k±0.5k** | 13.2k±0.3k |
| assault | **36.7k±3.0k** | 34.8k±5.4k | 1.9k±0.6k |
| hero | 54.5k±3.0k | **69.4k±5.7k** | 64.2k±7.0k |
| ice hockey | **85.1±0.6** | -4.1±0.3 | -0.5±0.2 |
| jamesbond | **30.8k±2.3k** | 26.6k±2.6k | 22.4k±0.3k |
| kangaroo | 14.7k±0.1k | **35.1k±2.1k** | 16.9k±2.0k |
| krull | **131.1k±12.7k** | 127.4k±17.1k | 26.7k±2.2k |
| kung fu master | **220.7k±3.9k** | 212.1k±11.2k | 203.2k±10.8k |
| montezuma revenge | 2.3k±0.4k | 10.4k±1.5k | **16.8k±6.8k** |
| ms pacman | 40.3k±1.1k | **40.8k±1.1k** | 38.1k±1.6k |
| name this game | **70.6k±8.3k** | 23.9k±0.5k | 15.6k±0.3k |
| phoenix | 935.2k±13.5k | **959.1k±2.7k** | 933.3k±4.7k |
| asterix | **994.3k±0.9k** | 950.7k±23.1k | 953.1k±21.7k |
| pitfall | -5.2±1.7 | **7.8k±4.0k** | 1.3k±0.9k |
| pong | **20.9±0.0** | 19.6±0.2 | -9.7±1.4 |
| private eye | 31.5k±5.4k | **100.0k±0.4k** | 65.6k±14.3k |
| qbert | 398.6k±46.4k | **451.9k±82.1k** | 449.0k±108.2k |
| riverraid | 38.9k±0.6k | 36.7k±2.3k | **42.6k±4.0k** |
| road runner | 105.2k±16.8k | **128.6k±28.7k** | 103.9k±7.1k |
| robotank | **142.4±0.8** | 9.1±0.7 | 24.9±2.5 |
| seaquest | **1000.0k±0.0k** | 1000.0k±0.0k | 1000.0k±0.0k |
| skiing | **-11081.7±97.7** | -22977.9±5059.9 | -21907.4±3647.5 |
| solaris | **5.6k±1.1k** | 4.7k±0.3k | 1.3k±0.1k |
| space invaders | 33.4k±12.2k | **43.4k±1.3k** | 12.0k±1.9k |
| star gunner | 412.8k±4.1k | 414.6k±66.8k | **452.5k±53.0k** |
| surround | **9.8±0.0** | -9.6±0.2 | -7.3±0.4 |
| tennis | **24.0±0.0** | 10.2±3.5 | -3.5±5.7 |

Table 10: Scores over all 57 Atari games.

# H NETWORK ARCHITECTURES

## H.1 ARCHITECTURE OF THE EMBEDDING NETWORK WITH INVERSE DYNAMICS PREDICTION

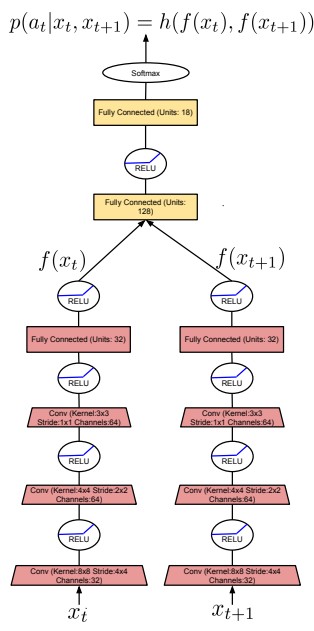

Figure 15: Embedding Network Architecture.

## H.2 ARCHITECTURE OF THE RANDOM NETWORK DISTILLATION

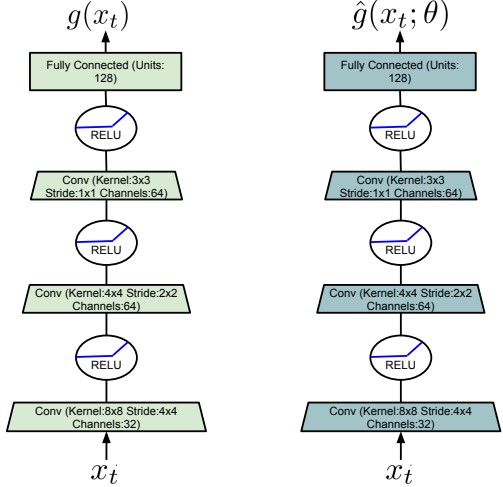

Figure 16: RND Network Architecture.

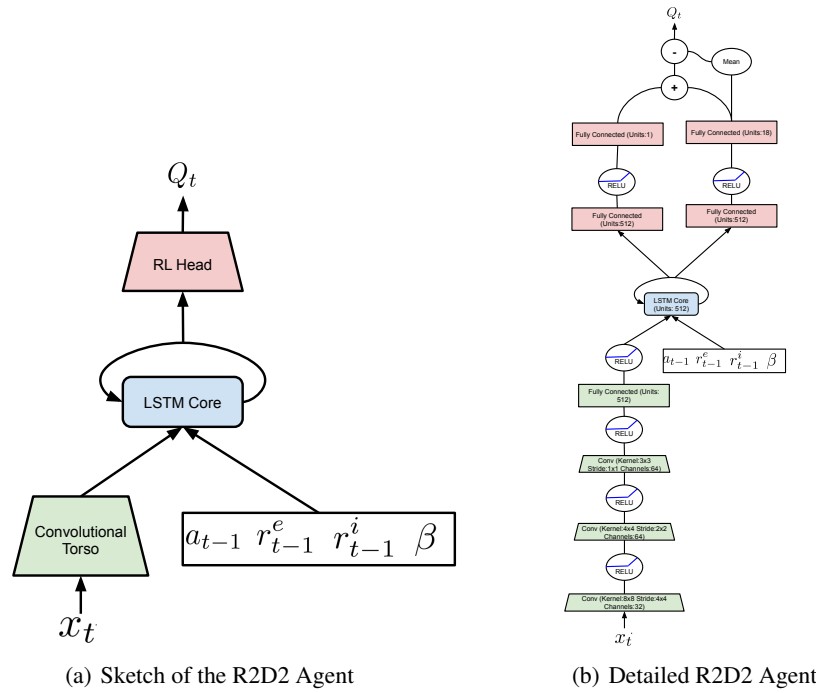

(a) Sketch of the R2D2 Agent    (b) Detailed R2D2 Agent

Figure 17: R2D2 Agent Architecture.

## H.3 ARCHITECTURE OF THE R2D2 AGENT

## I CONTROLLABLE STATES

In this section we evaluate properties of the learned controllable states. We further present a study of the performance of the algorithm when having access to oracle controllable states containing only the necessary information. We use Montezuma's Revenge as a case-study.

### I.1 INSPECTING THE PROPERTIES OF LEARNED CONTROLLABLE STATES

As explained in Section 2, we train the embedding network $f$ using an inverse dynamics model as done by Pathak et al. (2017). Intuitively, the controllable states should contain the information relevant to the action performed by the agent given two consecutive observations. However it might contain other type of information as long as it can be easily ignored by our simple classifier, $g$.

As noted in Burda et al. (2018b), for this game, one can identify a novel state by using five pieces of information: the $(x, y)$ position of the player, a room identifier, the level number, and the number of keys held. This information can be easily extracted from the RAM state of the game as described in Section I.3 bellow. One question that we could ask is whether this information is present (or easily decodable) or not in the learned controllable state. We attempted to answer this question by training a linear classifier to predict the $(x, y)$ coordinates and the room identified from the learned controllable state. Importantly we do not backpropagate the errors to the embedding network $f$. Figure 19 shows the average results over the episodes as the training of the agent progresses. We can see that the squared error in predicting the $(x, y)$ position of the agent stabilises to a more or less constant value, which suggests that it can successfully generalise to new rooms (we do not observe an increase in the error when new rooms are discovered). The magnitude of the error is of the order of 12 units, which less than 10% of the range (see Section I.3). This is to be expected, as it is the most important information for predicting which action was taken. It shows that the information is quite accessible and probably has a significant influence in the proposed novelty measure. The room identifier, on the other hand, is information that is not necesary to predict the action taken by the agent. Unlike the previous case, one can see jumps in the error as training progresses as the problem becomes harder.

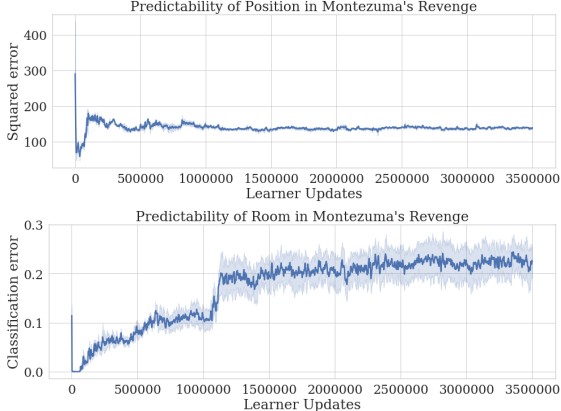

Figure 18: Prediction loss from learned controllable states of the position (top) and room (bottom) on Montezuma's Revenge against minibatch updates.

It stabilises around an error slightly above 20%, which is reasonably good considering that random chance is 96%. This means that even if there is nothing specifically encouraging this information to be there, it is still present and in turn can influence the proposed novelty signal.

An avenue of future work is to research alternative methods for learning controllable states that directly search for retaining all relevant information. While very good results can be obtained with one of the simple alternative of an inverse dynamics model, it is reasonable to think that better results could be attained when using a better crafted one. To inform this question, we investigate in the next section what results could we obtain if we explicitly use as controllable states the quantities that we were trying to predict in this section.

## I.2 MONTEZUMA'S REVENGE WITH HAND-CRAFTED CONTROLLABLE STATES

In the previous section we analysed the properties of the learned controllable states. A valid question to ask is: how would the NGU work if we had access to an oracle controllable state containing only the relevant information? This analysis is a form of upper bound performance for a given agent architecture. We ran the NGU(N=1) model with two ablations: without RND and without extrinsic rewards. Instead of resetting the memory after every episode, we do it after a small number of consecutive episodes, which we call a *meta-episode*. This structure plays an important role when the agent faces irreversible choices. In this setting, approaches using non-episodic exploration bonuses are even more susceptible to suffer from the "detachment" problem described in Ecoffet et al. (2019). The agent might switch between alternatives without having exhausted all learning opportunities, rendering choosing the initial option uninteresting from a novelty perspective. The episodic approach with a meta-episode of length one would be forced to make similar choices. However, when run with multiple episodes it can offer an interesting alternative. In the first episode, the agent starts with an empty episodic memory can can choose arbitrarily one of the options. In the second episode, the episodic memory contains all the experience collected in the first episode. The agent is then rewarded for *not* repeating the strategy followed in the first one, as revisiting those states will lead to lower intrinsic reward. Thus, the agent is encourage to learn diverse behaviour across episodes without needing to choose between alternatives nor being susceptible to the detachment problem. Results are summarized in Fig. 19. We report the average episodic return (left) as well as the average number of visited rooms per meta-episode (right). The model achieves higher scores than the one using learned controllable states (as reported in Section 4.2).

Incorporating long-term novelty in the exploration bonus, encourages the agent to concentrate in the less explored areas of the environment. Similarly to what we observed with learned controllable states, this provides a boost both in data efficiency as well as final performance, obtaining close to 15,000 average return and visiting an average of 25 rooms per episode. In this run, three out of five seeds reach the second level of the game, one of which reaches the third level with an average of fifty different rooms per episode. We also observe that, when running in the absence of extrinsic rewards, the agent remarkably still achieves a very high extrinsic reward. Secondly, the agent is able

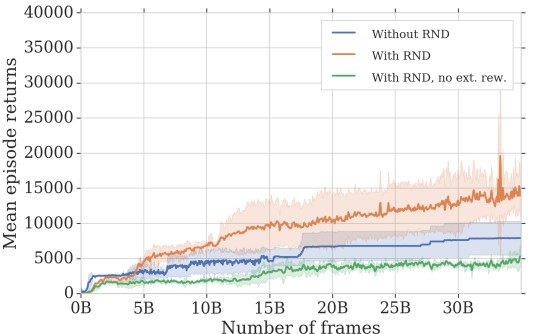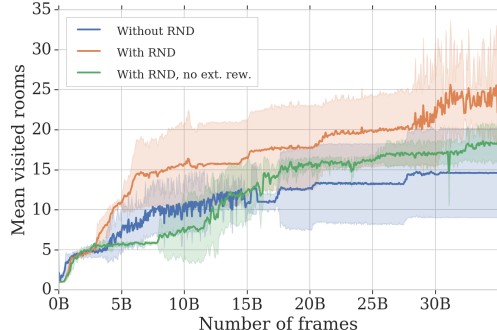

Figure 19: Mean episodic return (left) and mean number of visited rooms per episode (right) vs environment frames for agents trained Montezuma's Revenge with hand-crafted controllable states.

to consistently reach a large number of rooms and explore more than 20 rooms without any extrinsic guidance.

As noted in Burda et al. (2018b), in Montezuma's Revenge each level contains 6 doors and 4 keys. If the agent walks through a door holding a key, it receives a reward of 300 consuming the key in the process. In order to clear a level, the agent needs open two doors located just before the final room. During exploration, the agent needs to hold on to two keys to see what it could do with them later in the episode, sacrificing the immediate reward of opening more accessible doors. Any agent that acts almost greedily will struggle with what looks like a high level choice. With the right representation and using meta-episodes, our method can handle this problem in an interesting way. When the number of keys held is represented in the controllable state, the agent chooses a different key-door combination on each of the three episodes in which we do not wipe our episodic memory. At the end of training, in the first episode after wiping the episodic memory, our agent shows a score of $14,660 \pm 196$, while the third episode the agent shows a score of $34,040 \pm 9,835$, exploring on average over 30 rooms and consistently going to the second level[3]. The agent learns a complex exploratory policy spanning several episodes that can handle irreversible choices and overcome "distractor" rewards. We do not observe different key-door combinations across episodes when using learned controllable states. Presumably the signal of the number of held keys in the learned controllable states is not strong enough to treat them as sufficiently different.

The results describe in this section support the idea that significant gains can be obtained by improving the respresentation of the controllable states, suggesting that the study of learning better representations is an interesting line for future work. Recent works have explored ways of measuring novelty by learning controllable aspects of an environment (Kim et al., 2018; Warde-Farley et al., 2018), and we believe that some of these ideas could be also useful in this setting.

## I.3 Hand-crafted state features for Montezuma's Revenge

We obtain the hand-crafted features for Montezuma's Revenge by observing the RAM state of the game at every time step. More concretely:

- x and y can be observed at positions 0xAA and 0xAB respectively, represented by integers with a range of $[0, 153] \times [0, 122]$.

- Room id and level number can be found in positions 0x83 and 0xB9 respectively. We provide this information as a single integer to our agent in the form of $r_{id} + 24 * l_n$ where $r_{id} \in \{0, \ldots, 23\}$ is the room id, and $l_n$ is the level number.

- Byte 0xC1 is the player's inventory. We count the number of keys being held (and provide this information to the agent) by adding the bits $\{2, \ldots, 6\}$, which correspond to the binary slots for keys.

---

[3]See video of the three episodes at `https://sites.google.com/view/nguiclr2020`

