# OpenReview forum: "Never Give Up: Learning Directed Exploration Strategies"
_ICLR.cc/2020/Conference — Accept (Poster)_

### Official Review · AnonReviewer3 · 2019-10-22
**Official Blind Review #3**

**Rating:** 8

**Review:**

The paper proposes a novel intrinsic reward/curiosity metric that combines both episodic and “life-long” novelty. Essentially two competing pressures that push agents to explore as many novel states in a single rollout as possible and to explore as many states as possible as evenly as possible. The primary contribution here is the episodic novelty measure, which relies on a state embedding that takes into account stochasticity in the environment. The paper covers this episodic curiosity measure and how it’s integrated with the life-long curiosity metric. It then demonstrates the impact of these metrics and variations compared to baselines on particular games and all 57 Arcade Learning Environment games.

This is a clear accept. This paper demonstrates a novel episodic curiosity metric and a means of integrating that with a more standard life-long curiosity metric. The writing is clear and the results are good and well-explained.

I would appreciate in the final paper some discussion of whether it would be possible to adjust the hyper-parameters of the approach during training, given that different variations of the approach seemed to do consistently better or worse as the authors described. Further, I would have appreciated a summarization of the limitations towards the end of the paper.

I recognize that the life-long curiosity approach is fairly arbitrary, but given that the controllable state is already available, I’m not sure why it isn’t used for this measure. It seems naively it would be helpful. If not, some clarity on this would be appreciated.

**Experience Assessment:**

I have read many papers in this area.

**Review Assessment: Checking Correctness Of Derivations And Theory:**

I did not assess the derivations or theory.

**Review Assessment: Checking Correctness Of Experiments:**

I assessed the sensibility of the experiments.

**Review Assessment: Thoroughness In Paper Reading:**

I read the paper at least twice and used my best judgement in assessing the paper.

---

> ### Author Response · Authors · 2019-11-12
> **Response to blind review #3**
>
> Thank you for your review. We believe the raised points greatly resonate with us, and they contribute to a greater overview and summarization of the method. More concretely:
>
> * Learning to adjust the hyper-parameters is something that we contemplate as a highly promising future direction for this work. More concretely, we think it should be possible to adapt the hyperparameter beta, which determines the maximum degree of exploration done, so that the method is able to dynamically adapt it, given that some games require much more exploration than others. Potential approaches are the use of techniques such as Population Based Training (PBT) or the use of Meta-gradients. Another advantage of such method would be to allow for the model to "turn off" exploration when the agent has reached a point in which further exploring does not lead to improvements on the exploitative policy. Having said that, including such a mechanism would require calibrating the adaptation to be aligned to the speed of learning of the exploitative policy.  We have included a discussion on this in the conclusions section of the updated paper.
>
> * We agree that a summarization of the limitations of the method is a useful addition to the conclusions of this work. In the updated version of the manuscript we have added a description of these limitations.
>
> *We agree with the reviewer that the controllable state could be used to determine long-term novelty. A natural first choice would be to follow the work in Curiosity-driven exploration and learn a forward model in this embedding space, and then use the prediction errors as an intrinsic motivation signal. While this is a good choice (and would be interesting to test), we decided to go with the RND variant as it avoids the difficulties of using forward prediction as novelty signals (the ‘noisy TV’ problem, as highlighted in the RND work), it has been shown to perform better empirically, and is amenable to distributed training. Another alternative would be to use the same idea as in the episodic novelty, but without emptying the memory at the end of an episode. This would naturally provide a longer term novelty signal. The clear limitations of that approach come from an implementation standpoint: it would require to store and perform look-ups over much larger episodic memories.

---

> > ### Comment · AnonReviewer3 · 2019-11-14
> > **Reviewer #3 response to author response**
> >
> > Thank you for the clarifying points and discussion. I have not updated my review given I already rated the paper as accept, but I appreciate the time put in to this response.

---

### Official Review · AnonReviewer1 · 2019-10-22
**Official Blind Review #1**

**Rating:** 6

**Review:**

-after rebuttal:
I read the replies from the authors and re-read the modified version of the paper and I believe there has been a noticeable improvement in the presentation. I still think it could be improved more (in terms of wording and better exposition of the results) but due to the in-place improvements I increase my score to weak accept.
----------------------------------------------------------------------------------------------------------------------------------------------------
In this paper, the authors present a methodology for generating intrinsic rewards for reinforcement learning agents targeting hard exploration environments. The intrinsic reward is generated using an episodic memory module and a lifelong novelty module.  A state representation is learnt in such a way that the novelty signals are biased towards what the agent can control. A single neural network learns the q-values of exploratory policies with different degrees of exploration. Several experiments on a simple domain and on Atari domains are conducted to evaluate and compare the performance of the proposed method against the baselines.


In my opinion, this is an interesting idea as it present a novel combination of methods (Random Network Distillation and Episodic Memory) that works and can inspire other researchers in the field.  However, I would like to see clearer explanations in the experimental section before acceptance. For this reason I rate this paper as a weak reject but if clarity is improved I will increase my score.  See below for more general comments and detailed explanation about the experimental section.

General comments:

- The authors state that a novel contribution is to disentangle exploration and exploitation. This is not true: see [1] for a recent paper on the topic. I believe the authors should cite this paper.

- In page 3: “To determine the bonus, the current observation is compared to the content of the episodic memory. Larger differences produce larger episodic intrinsic rewards” When computing the intrinsic reward for a new state, it must be compared to the episodic memory, which is composed by recent states (and old). Therefore, in this situation the intrinsic reward is always high? Specially because it is compared with the k-nearest neighbours?  Maybe the authors could elaborate on this?



Experiment Section:

This section should be tidied up in my view.  In general, I feel that there are too many fine-grained results / experiments. I think some aggregation of results would be good for clarity. Without this aggregation there are statements made from the authors that are difficult to believe, e.g. a general result statement is valid for one or two games but not for the rest (but still the statement is formulated in a general way). This can lead to misunderstandings and overstatements.  This together with lack of some experimental details makes the section very dense and difficult to parse. Below I make my points. They are in order of appearance in the main manuscript. I marked with (*) the ones I consider the most important.

Section 4.1:

- Why the exploration policy in this section is set to have $\beta = 0.3$ ? Is my understanding correct that any value of $\beta$ would produce similar results since this is just a scaling of the intrinsic reward (i.e. it does not balance exploration vs exploitation)?

- The parameter $\beta$ appears in the construction of the reward r = r^e + \beta r^i , but also the output of Algorithm 1 scales the similarity by $\beta$ does that mean that the final contribution of $\beta$ is squared? Is this intended? If so, why?

- At the end of this section: “However, staying still is enough: staying still every state will produce ….” Since the agent can only take 4 action {left, right, up, down}, what do the authors mean by “staying still”, is really the agent doing some sort of cyclic policy e.g. left right left right … ?

Section 4.2:

- In the appendix A it is stated “use last 5 frames of the sampled sequences to train the action prediction network… “ Does this refer to frame-stacking? I assume it is not since at the beginning section 4.2 it is stated that there is no frame-stacking. If it is not frame-stacking, the authors could explain in more detail what do they refer to.

- In the paragraph “Architecture” it is stated that 8 games were selected to choose the hyperparemeters and that the results are in Appendix B. However, appendix B only shows 2 games (Pitfall and Montezuma’s Revenge). Is this a typo?

- (*) In paragraph “NGU Agent”: This is the most dense paragraph and the most difficult to parse. First of all Table 1 shows all the results, but as one can see, the different ablations have very similar performance on most games with only a few exceptions. Note that most of the mean performances with error bars, have actually overlapping error bars for many combinations of games and methods. Therefore, further statements about this table are difficult to believe since they could have been just the result of random seeds. I think it is fine to show these fine-grained results on the appendix, but I would say it would be better to aggregate them in the main paper and show that the statements made by the authors still hold for this aggregation.

Specific comments about NGU agent paragraph:

- “ we observe an improvement from increasing the number of mixtures (with diminishing returns) on hard exploration games.” I would say the authors cannot claim this since the hard exploration games are the ones in table 2, which are different from the ones in table 1 (only Pitfall, MR and Private Eye coincide). Also the statement is only true for 2 the three that are hard exploration games (Pitfall and Private Eye).

- (*)  “for smaller $\beta$ we observe better performance on Pong and Beam Rider, but worse performance on all hard exploration games” This is a strange result in my opinion. If I understood correctly, the base agent has $\beta = 0.3$, which supposedly has been selected to be good on hard-exploration games. However, here only changing $\beta$ slightly, reduces the performance on hard-exploration games (Pitfall and Private Eye) significantly. Is this due to the parameter being very sensitive? If so, I believe the authors should report how sensitive are the results to this parameter, specially, on the full hard-exploration games.

- “superhuman performance on 3 games”: which ones?

Comments on paragraph on “hard exploration games”:
- “with NGU(N=1)-RND ….” what do the authors mean by this? This seems to be the best setting for Pitfall but it is actually not using mixture of explorations and (I guess) neither RND?
- Table 2 first row “best base” what does this mean?


Minor Comments:

-In Equation 3 the squared distance is normalized by a running average. Why?

- Right after Equation 3: “… episodic reward can be found in Alg. 14” . Probably meant Alg 1.

- Similarly why the errors are normalized by a running average when computing \alpha_t ?


[1] MULEX: Disentangling Exploitation from Exploration in Deep RL ( Lucas Beyer et al )



**Experience Assessment:**

I have published one or two papers in this area.

**Review Assessment: Checking Correctness Of Derivations And Theory:**

I assessed the sensibility of the derivations and theory.

**Review Assessment: Checking Correctness Of Experiments:**

I carefully checked the experiments.

**Review Assessment: Thoroughness In Paper Reading:**

I read the paper thoroughly.

---

> ### Author Response · Authors · 2019-11-12
> **Response to official blind review #1 part 1**
>
> Thank you for your greatly thorough review. We believe that this thoroughness (including careful attention to the Appendix) and valuable questions have strongly contributed to improving the clarity of the paper, specially regarding the clarity of the experiments section of the paper. We have incorporated the very helpful feedback from the reviewer and updated the manuscript accordingly. We hope that this is enough for the reviewer to revisit their score of our manuscript. We proceed to answer in order:
>
> General comments:
> * Concerning the contribution of disentangling exploration and exploitation we should have been more careful in our claims and we are aware that other methods exist in the literature. Rather than a novel contribution, it is an interesting property of our approach that we wanted to highlight, and contrasts with most recent work in intrinsic motivation. We have amended the wording to reflect this.
>
> MULEX is indeed very related work, and we were not aware of it at the time of writing this paper as it was posted on arXiv very recently and is, as far as we can tell, unpublished. We thank the reviewer for pointing this out. In the updated version of the manuscript we include the suggested citation and describe its relation with NGU after describing our contributions. The main difference between the way this work and MULEX combine the exploitation and exploration is that our approach does it by sharing weights between the different policies, contrary to MULEX, which shares data in a common replay buffer. One could argue that imitation learning is also a way to disentangle exploration and exploitation, done by sharing human demonstrations in the replay buffer (DQfD method). In the case of MULEX, the data added to the replay does not come from human demonstrations but by trained exploration policies.
>
> * This statement in our work was not very clear. When computing the intrinsic reward for a new state, we indeed compare it to the content of the episodic memory. We meant to say that, the more dissimilar the past controllable states are from the new state, the higher the reward. Looking at equation (2) in the manuscript, this means that when the new state is very different from the content of the memory, the term \sqrt{\sum_{f_i\in N_k} K(f(x_t), f_i)} in the denominator will be small, and thus the reward will be high. Hence, in order to maximise the episodic intrinsic reward, the agent needs to go to states that have low similarity with the previously visited ones, consequently encouraging exploration.
>
> In practice (2) is computed using the k-nearest neighbors from memory. If the new state is dissimilar to the k-nearest memories, it will be more dissimilar from the remaining ones. Using only k-nearest neighbors allows for faster computations and is common practice in methods using content-based look-ups on episodic memories (e.g. Neural Episodic Control)
>
> Experiments section:
> Section 4.1:
> - Yes, that is correct in that beta=0.3  does not precisely balance exploration and exploitation. Let us consider the case of a single policy (N=1) with a given fixed value of beta. This policy is trained using the augmented reward r=r_e+beta r_i. Thus, the larger the value of beta, the less important the extrinsic rewards will be relative to the intrinsic rewards. Thus, there is an implicit exploitation-exploration trade off within the definition of the exploratory policy.
> In the case with multiple mixtures, the value of beta represents the weight of the intrinsic reward for the most exploratory policy. Here again, the higher the beta, the more exploratory the policy will be. As seen in Table 1, different values of beta (beta = 0.2, beta=0.3, beta = 0.5) lead to different results (see extended discussion on our other comment below regarding the sensitivity w.r.t. this hyperparameter).
>
>
> - This is not intended, thank you for catching this. We have updated the paper with the correction. Beta is only multiplied once, so we have removed it from the algorithm.
> - Yes, that is correct. We have changed the wording to reflect this.
>
> Section 4.2:
> - Yes, it is not frame-stacking. This refers to the data received by the learner in R2D2. In R2D2, a learner samples and learns from a batch of trajectories coming from a set of actors running in parallel on separate environment instances. In our case, the length of each sequence in the batch is of 80 time steps (with minibatch size of 64 this means 80x64=5120 different states). We use all the data to train the RL loss. In contrast, we use a subset of the data to train the action prediction and the RND losses. Training these networks on 5120 states per batch is computationally inefficient. To select a subset of the data we simply take the last 5 time steps of each sequence in the batch, as we saw that to be empirically enough in our original experimentation (similarly done in the RND work). We have changed our description in the appendix to clarify this.

---

> ### Author Response · Authors · 2019-11-12
> **Response to official blind review #1 part 2**
>
> Please see part 1 first.
> - Yes, it is a typo. We selected the hyperparameters that relate to exploration based on the 2 games of appendix B, and not the 8 games in which we do the ablations of the hyperparameters of general effect. We corrected this and updated the paper with the correction.
>
> - (*) We agree this was a very dense paragraph. We have changed Table 1 to a Figure with aggregated results, and we have moved the original table to the appendix. Thanks for this very helpful suggestion, we agree it improves the presentation of the ablations. We have reworded this paragraph to make it easier to parse. Whilst the standard deviation error bars do indeed overlap, the means do consistently shift on hard exploration games and superhuman performance is most reliably obtained using the full combination.
>
> Specific comments about NGU agent paragraph:
> - Thank you for catching this. It is true we cannot claim a progression with diminishing returns (although there is a slight trend), and further we were not clarifying that this only refers to the average performance over the 3 hard exploration games listed on Table 1. We have modified our claim which we believe that is now more fair and better adjusted to the empirical results that we observe.
>
> - (*)  Beta is a hyperparameter that affects the exploration/exploitation trade-off. Tuning it will change the performance of the agent, as evidenced in the ablations Table of the Appendix. A lower beta yields less exploration and so it is not surprising that reducing it yields to better performance on games not requiring extensive exploration (Pong and Beam rider). With regards to sensitivity, it is important to note that we tuned beta to give the best performance on 5 dense and 3 hard exploration games then demonstrated that the selected value generalised well to perform well on all 57 Atari games. beta does not require per-game tuning to yield reasonably good performance.
> That said, further tuning beta can certainly lead to improvements, and this is illustrated in the ablations Table. But note that, for all positive beta values tested, achieve non-zero reward on Pitfall, while in private eye all variants still outperform all other baselines we compare to. Specifically, in the case of Private eye the distance in score might be misleading, as rewards are very sparse of large value. For instance, after reaching a score of 40k, (ignoring some smaller rewards that add up to less than 2000 points) there are only two rewards to be collected of around 30k points. This creates what seems to be large differences in the scores. We have added this important clarification on the performance on hard-exploration games in our analysis of the ablations Table in the appendix. Finally, we have also added a section on the performance of beta = 0.2 and beta = 0.5 to that appendix analysis. As one can see, the games in which the beta parameter is most sensitive are the ones that were already shown on Table 1. We agree with Reviewer 3 that a natural extension of this model is to find ways of dynamically adjusting this hyperparameter in an online manner (please see answer to reviewer 3 below).
>
> - This refers to Breakout, Space Invaders, and QBert. Given that based on the above comment we have changed Table 1 to be more compact figure, we clarified this conclusion in the ablations Table of the appendix. It may also be difficult to compare since the human baselines are in other tables, therefore we have added a human baseline row to the table of ablations for NGU(N=32) of the appendix.
>
> Comments on paragraph on “hard exploration games”:
> - Yes that is correct, NGU(N=1)-RND means training a single policy and without the use of the RND reward. This setting achieves the highest score for Pitfall. Our intuition is that in this case a single policy can achieve quite good results since exploration and exploitation policies are similar. As far as RND usage is concerned, we consistently observed that not using RND on Pitfall! leads to results that are qualitatively similar, but much more data efficient. We bring attention to the graphs shown on figure 4, and in our analysis we highlight 3 hypotheses explaining why this may happen.
> We have modified our analysis to reflect these comments.
>
>
> - It refers to the ‘best baseline’ described in the table description. To clarify this, we have changed the name of the row to use the full name instead of this abbreviation.
>
> Minor Comments:
>
> - This is to make the kernel more robust to the task being solved, as different games may have different typical distances between learnt embeddings. We have added a note to clarify this.
>
> - Yes, we have corrected this.
>
> - This is done in the original implementation of the RND reward. It shares motivation with the answer to the first minor comment. The RND reward is described as being normalized by a running average and standard deviation of the rewards. We have changed our wording of the definition of \alpha_t to reflect this.

---

### Official Review · AnonReviewer2 · 2019-10-22
**Official Blind Review #2**

**Rating:** 6

**Review:**

The work is motivated by the goal of having a comprehensive exploration of an agent in deep RL. For achieving that, the authors propose a count-based NGU agent, combining intrinsic and extrinsic bonuses as new rewards. An extrinsic/ long-term novelty module is used to control the amount of exploration across episodes, a life-long curiosity factor as its output. In the intrinsic/episodic novelty module, an embedding net and a KNN on episodic memory are applied to compute the current episodic reward. In the experiment, a universal value function approximator (UVFA) framework is used to simultaneously approximate the optimal value function with a set of rewards. The proposed method is tested on several hard exploration games. Other recent count-based models are compared in the paper.

Cons:
- To my acknowledge, the task and the count-based methods are not too novel.
- They use 35 billion environment frames.

Overall, this paper is well-written. Methods and results are clearly described.


**Experience Assessment:**

I do not know much about this area.

**Review Assessment: Checking Correctness Of Derivations And Theory:**

I assessed the sensibility of the derivations and theory.

**Review Assessment: Checking Correctness Of Experiments:**

I assessed the sensibility of the experiments.

**Review Assessment: Thoroughness In Paper Reading:**

I read the paper at least twice and used my best judgement in assessing the paper.

---

> ### Author Response · Authors · 2019-11-12
> **Response to official blind review #2**
>
> Thank you for review. We appreciate the concerns raised and we wish to provide further clarity on those points. More concretely:
>
> We agree that there is already extensive literature on count-based methods. In our view, while a lot of progress has been made in recent years (covered by the work we cite in the manuscript),  the problem of extending count-based exploration methods to very large high dimensional state spaces (the usual setting in deep RL) is not yet solved. Given this body of literature, our method contributes 1) an exploration bonus we define, which combines life-long novelty and episodic novelty, 2) learning a family of policies that separate exploration and exploitation with shared weights, and 3) strong experimental results, with State of the Art on games such as Pitfall, where no algorithm (without demonstrations or privileged information) was performing better than random.
>
> Current state of the art methods in deep RL achieve their results by leveraging large amounts of compute by running on distributed training architectures that collect large amounts of experience from many actors running in parallel on separate environment instances.  This is also the case of R2D2, the state-of-the-art agent on the Atari suite. Despite having an incredibly high average (or median) performance, it performs poorly on most hard exploration games. We agree with the reviewer that an important line for future research is to explore effective ways of significantly improving NGU's data efficiency while maintaining its performance.
>
> Having said that, we want to stress that the goal of this work is to push the limits of the best performing agents available in the literature. This is, we want to push the limits of performance: what are the best achievable scores when data or compute are not a limitation?
>
> Finally, we want to point out that running for 35 billion frames is not used to NGU's advantage. All the baselines we implemented are also run for that number of steps. This includes R2D2 + RND, which obtains slightly stronger results than the original RND publication, but much weaker than the results obtained with NGU. In summary, unlike the baselines we compare to that we implemented, with the same amount of frames consumed, our method is able to reliably leverage that amount of compute to achieve better final performance. A comparison of the computation used by all the baselines is described in Appendix C.

---

### Decision · Program_Chairs · 2019-12-19

**Decision:**

Accept (Poster)

**Comment:**

This paper tackles hard-exploration RL problems. The idea is to learn separate exploration and exploitation strategies using the same network (representation). The exploration is driven by intrinsic rewards, which are generated using an episodic memory and a lifelong novelty modules. Several experiments (simple and Atari domains) show that the proposed approach compares favourably with the baselines.

The work is novel both in terms of the episodic curiosity metric and its integration with the life-long curiosity metric, and the results are convincing. All reviewers being positive about this paper, I therefore recommend acceptance.